# An intercomparison of CH₃O₂ measurements by Fluorescence Assay by Gas Expansion and Cavity Ring–Down Spectroscopy within HIRAC (Highly Instrumented Reactor for Atmospheric Chemistry)

Lavinia Onel[1], Alexander Brennan[1], Michele Gianella[2], James Hooper[1], Nicole Ng[2], Gus Hancock[2], Lisa Whalley[1,3], Paul W. Seakins[1], Grant A. D. Ritchie[2], Dwayne E. Heard[1]

[1] School of Chemistry, University of Leeds, Leeds, LS2 9JT, UK
[2] Department of Chemistry, Physical and Theoretical Chemistry Laboratory, University of Oxford, Oxford, OX1 3QZ, UK
[3] National Centre for Atmospheric Science, University of Leeds, Leeds, LS2 9JT, UK

*Correspondence to*: Lavinia Onel (chmlo@leeds.ac.uk); Paul Seakins (p.w.seakins@leeds.ac.uk); Grant Ritchie (grant.ritchie@chem.ox.ac.uk); Dwayne Heard (d.e.heard@leeds.ac.uk)

**Abstract**

Simultaneous measurements of CH₃O₂ radical concentrations have been performed using two different methods in the Leeds HIRAC (Highly Instrumented Reactor for Atmospheric Chemistry) chamber at 295 K and in 80 mbar of a mixture of 3:1 He:O₂ and 100 mbar or 1000 mbar of synthetic air. The first detection method consisted of the indirect detection of CH₃O₂ using the conversion of CH₃O₂ into CH₃O by excess NO with subsequent detection of CH₃O by fluorescence assay by gas expansion (FAGE). The FAGE instrument was calibrated for CH₃O₂ in two ways. In the first method, a known concentration of CH₃O₂ was generated using the 185 nm photolysis of water vapour in synthetic air at atmospheric pressure followed by the conversion of the generated OH radicals to CH₃O₂ by reaction with CH₄/O₂. This calibration can be used for experiments performed in HIRAC at 1000 mbar in air. In the second method, calibration was achieved by generating a near steady-state of CH₃O₂ and then switching off the photolysis lamps within HIRAC and monitoring the subsequent decay of CH₃O₂ which was controlled via its self-reaction, and analysing the decay using second order kinetics. This calibration could be used for experiments performed at all pressures. In the second detection method, CH₃O₂ has been measured directly using Cavity Ring-Down Spectroscopy (CRDS) using the absorption at 7487.98 cm⁻¹ in the $A \leftarrow X$ (ν₁₂) band with the optical path along the ~1.4 m chamber diameter. Analysis of the second-order kinetic decays of CH₃O₂ by self-reaction monitored by CRDS has been used for the determination of the CH₃O₂ absorption cross section at 7487.98 cm⁻¹, both at 100 mbar of air and at 80 mbar of a 3:1 He:O₂ mixture, from which $\sigma_{CH3O2} = (1.49 \pm 0.19) \times 10^{-20}$ cm² molecule⁻¹ was determined for both pressures. The absorption spectrum of CH₃O₂ between 7486 and 7491 cm⁻¹ did not change shape when the total pressure was increased to 1000 mbar, from which we determined that $\sigma_{CH3O2}$ is independent of pressure over the pressure range 100–1000 mbar in air. CH₃O₂ was generated in HIRAC using either the photolysis of Cl₂ with UV black lamps in the presence of CH₄ and O₂ or the photolysis of acetone at 254 nm in the presence of O₂. At 1000 mbar of synthetic air the correlation plot of [CH₃O₂]_FAGE against [CH₃O₂]_CRDS gave a gradient of $1.09 \pm 0.06$. At 100 mbar of synthetic air the FAGE – CRDS correlation plot had a gradient of $0.95 \pm 0.02$ and at 80 mbar of 3:1 He:O₂ mixture the correlation plot gradient was $1.03 \pm 0.05$. These results provide a validation of the FAGE method to determine concentrations of CH₃O₂.

## 1 Introduction

Methyl peroxy (CH₃O₂) radicals are important intermediates during atmospheric oxidation (Orlando and Tyndall, 2012) and combustion chemistry (Zador et al., 2011), and are produced mainly by the oxidation of CH₄ and larger hydrocarbons followed by the termolecular reaction between the CH₃ radical, O₂ and a third body (Reaction R1).

$$CH_3 + O_2 + M \rightarrow CH_3O_2 + M \qquad\qquad (R1)$$

In environments influenced by anthropogenic $NO_x$ emissions, $CH_3O_2$ predominantly reacts with NO to produce $NO_2$ and $CH_3O$ (Reaction R2).

$$CH_3O_2 + NO \rightarrow CH_3O + NO_2 \tag{R2}$$

$CH_3O$ subsequently reacts with $O_2$ (Reaction R3) to generate $HO_2$, which in turn oxidises another NO molecule to $NO_2$ (Reaction R4). The subsequent photolysis of $NO_2$ leads to the formation of tropospheric ozone, an important constituent of photochemical smog.

$$CH_3O + O_2 \rightarrow CH_2O + HO_2 \tag{R3}$$
$$HO_2 + NO \rightarrow OH + NO_2 \tag{R4}$$

In remote, clean environments, i.e. under low $NO_x$ levels, $CH_3O_2$ is significantly removed by its self-reaction (Reaction R5)
and the cross-reactions with $HO_2$ and other organic peroxy radicals ($RO_2$) (Tyndall et al., 2001).

$$CH_3O_2 + CH_3O_2 \rightarrow CH_3OH + CH_2O + O_2 \tag{R5a}$$
$$CH_3O_2 + CH_3O_2 \rightarrow CH_3O + CH_3O + O_2 \tag{R5b}$$

Recently the reaction of $CH_3O_2$ with OH was measured to be fast (Fittschen, 2019;Yan, 2016) and provides an additional loss route for $CH_3O_2$ under low $NO_x$ conditions (Fittschen et al., 2014;Assaf et al., 2017). As $CH_3O_2$ is formed by the oxidation of $CH_4$, one of the most abundant tropospheric trace gases, as well as by the oxidation of other volatile organic compounds, it is predicted by numerical models to be the most abundant $RO_2$ species in the atmosphere. Although $CH_3O_2$ has not (yet) been selectively measured in the atmosphere, its concentration has been estimated using atmospheric models to peak at $\sim 10^7 - 10^8$
molecule $cm^{-3}$ during the daytime (Whalley et al., 2010;Whalley et al., 2011;Whalley et al., 2018).

At present, $CH_3O_2$ is not measured selectively in the atmosphere by any direct or indirect method. The sum of $HO_2$ and all $RO_2$ species, $[HO_2] + \sum_i [RO_{2,i}]$, and separately, the sum of $RO_2$, $\sum_i [RO_{2,i}]$, have been measured in the atmosphere using a range of indirect methods. Onel et al. (2017a) presents an overview of these methods, such as the peroxy radical chemical amplifier (PERCA) (Cantrell et al., 1984;Hernandez et al., 2001;Green et al., 2006;Miyazaki et al., 2010;Wood et al., 2017),
$RO_x$ chemical conversion – CIMS (chemical ionisation mass spectrometry) ($RO_x$MAS) (Hanke et al., 2002) and $RO_x$ chemical conversion – LIF (laser induced fluorescence) ($RO_x$LIF) (Fuchs et al., 2008;Whalley et al., 2013). $RO_x$LIF uses LIF detection of OH at low pressure, known as fluorescence assay by gas expansion (FAGE) and has been employed for partially speciated $RO_2$ detection, distinguishing between the sum of alkene, aromatic and long-chain alkane-derived $RO_2$ radicals and the sum of short-chain alkane-derived $RO_2$ radicals (Whalley et al., 2013;Whalley et al., 2018).
CIMS methods using reagent ions such as $H_3O^+(H_2O)n$, $NO_3^-$ and $NH_4^+$ have been employed in the simultaneous and selective detection of $RO_2$ in a number of recent studies (Noziere and Hanson, 2017;Noziere and Vereecken, 2019;Hansel et al., 2018;Jokinen et al., 2014). Volatile small $RO_2$ radicals such as $CH_3O_2$ have been selectively measured in CIMS laboratory experiments with detection limits between $\sim 1 \times 10^8 - 1 \times 10^9$ molecule $cm^{-3}$ (Noziere and Hanson, 2017). CIMS with $NO_3^-$ reagent ion has been employed in field measurements to record diurnal profiles of some highly oxygenated low–vapour
pressure $RO_2$ radicals produced in the ozonolysis of monoterpenes peaking at a few $10^7$ molecule $cm^{-3}$ (Jokinen et al., 2014).

Many of the early laboratory studies of the $CH_3O_2$ radical reactions employed UV-absorption spectroscopy to monitor the $B \leftarrow X$ band centred around 240 nm, that is common to alkyl $RO_2$ species (Wallington et al., 1992;Tyndall et al., 2001). The similarity of the broad featureless UV-absorption spectra of $RO_2$ radicals made it challenging to distinguish between the

individual RO$_2$ species, particularly in a mixture (Orlando and Tyndall, 2012). The sensitivity of UV-absorption spectroscopy is quite low, for example a minimum detectable absorption of $5 \times 10^{-3}$, corresponding to $4 \times 10^{12}$ molecule cm$^{-3}$ CH$_3$O$_2$ was reported (Sander and Watson, 1980). The $A \leftarrow X$ electronic transition of RO$_2$ in the near IR (NIR) displays more structured spectra than the UV region, allowing a selective identification of RO$_2$ radicals. However the $A \leftarrow X$ transition is weaker than

the $B \leftarrow X$ transition and multipass arrangements have been used to improve the detection sensitivity. A step-scan Fourier Transform Infrared spectrometer (Huang et al., 2007) operated using a multipass White cell has been used to detect a number of RO$_2$ species, including CH$_3$O$_2$, with a typical minimum detectable absorbance of $\sim 1 \times 10^{-4}$, corresponding to a limit of detection (*LOD*) of $\sim 1 \times 10^{13}$ molecule cm$^{-3}$ for most RO$_2$ species studied. The use of cavity ring-down spectroscopy (CRDS) further improves the sensitivity of the RO$_2$ detection due to the significantly longer pathlengths that can be realized and to the

coupling of high performance NIR lasers, detectors and optical components. For example, an absorbance detection limit of less than $1 \times 10^{-6}$ has been obtained by using cavity mirrors of a maximum reflectivity of 99.995% (Atkinson and Spillman, 2002).

The CRDS technique has been used under both ambient and jet-cooled conditions to provide insight into the molecular structure of CH$_3$O$_2$ and more complex RO$_2$, and to selectively measure [RO$_2$] in the laboratory (Sharp et al., 2008;Kline and

Miller, 2014;Pushkarsky et al., 2000;Farago et al., 2013;Atkinson and Spillman, 2002;Sprague et al., 2013). Good agreement has been found between the experimental spectrum of CH$_3$O$_2$ in the range between $\sim$7200–8600 cm$^{-1}$ ($\sim$1.18–1.40 μm) measured using pulsed CRDS at typically 200 mbar of N$_2$:O$_2$ = 1.5:1.0 and theoretical predictions (Chung et al., 2007;Sharp et al., 2008). The origin band of the $A \leftarrow X$ transition has been located at 7382.8 cm$^{-1}$ and a value of $2.7 \times 10^{-20}$ cm$^2$ molecule$^{-1}$ has been estimated for the absorption cross section at this wavenumber (Pushkarsky et al., 2000;Chung et al., 2007). A weaker

absorption band has been found at 7488 cm$^{-1}$ and assigned to a transition involving the methyl torsion ($\nu_{12}$) (Pushkarsky et al., 2000;Chung et al., 2007). By using the CH$_3$O$_2$ spectrum measured by Pushkarsky et al. (2000) from 7300–7700 cm$^{-1}$, which covers both the origin band and the band involving the methyl torsional mode, a value of ca. $1.0 \times 10^{-20}$ cm$^2$ molecule$^{-1}$ is estimated for the maximum cross section for the $\nu_{12}$ transition, $\sigma_{max}(\nu_{12})$. A few years later, (Atkinson and Spillman, 2002) measured $\sigma_{max}(\nu_{12}) = (1.5 \pm 0.8) \times 10^{-20}$ cm$^2$ molecule$^{-1}$ at 27 mbar N$_2$:O$_2$ = 4:1 using continuous-wave (cw) CRDS. Very

recent cw-CRDS studies reported $\sigma_{max}(\nu_{12}) = 2.2 \times 10^{-20}$ cm$^2$ molecule$^{-1}$ at 67 mbar of a He + O$_2$ mixture (Fittschen, 2019) and no dependence of $\sigma_{max}(\nu_{12})$ on pressure over the range from 67 to 133 mbar (Farago et al., 2013).

Recently we have developed a new method for the selective and sensitive detection of CH$_3$O$_2$ using the conversion of CH$_3$O$_2$ to CH$_3$O with excess NO followed by CH$_3$O detection by FAGE with laser excitation at *ca*. 298 nm (Onel et al., 2017b). The *LOD* for the method whilst sampling from atmospheric pressure is $\sim 4.0 \times 10^8$ molecule cm$^{-3}$ for a signal-to-noise ratio of

2 and 5 min averaging time; the *LOD* is reduced to $\sim 1.0 \times 10^8$ molecule cm$^{-3}$ by averaging over 1 hour. Therefore, the method has potential to be used in the measurement of atmospheric levels of CH$_3$O$_2$ in clean environments where [CH$_3$O$_2$] has been calculated to be a few $10^8$ molecule cm$^{-3}$ (Whalley et al., 2010;Whalley et al., 2011). As LIF is not an absolute method of detection, FAGE instruments require calibration. Two methods of calibration for CH$_3$O$_2$ have been used (Onel et al., 2017b): the 184.9 nm photolysis of water vapour in the presence of excess CH$_4$ and the kinetics of the second-order decay of CH$_3$O$_2$

via its self–reaction observed in the Highly Instrumented Reactor for Atmospheric Chemistry (HIRAC). Good agreement was found, *i.e.* the calibration factors obtained using the two methods had overlapping error limits at the $1\sigma$ level.

However, radicals are difficult to detect accurately and, particularly as FAGE is not an absolute and direct method, may be subject to systematic errors and, hence require validation using complementary methods. Recently we intercompared measurements of HO$_2$ concentrations by the indirect FAGE method and the direct and absolute CRDS method within HIRAC,

and demonstrated good agreement, within 10% and 16% at 150 mbar and 1000 mbar, respectively (Onel et al., 2017b), which validates the FAGE method for HO$_2$. In this work, CH$_3$O$_2$ measurements by FAGE and CRDS within HIRAC are intercompared at 80 mbar for a mixture of 3:1 He:O$_2$ and at 100 mbar and 1000 mbar for air.

## 2 Experimental

### 2.1 CH₃O₂ generation in HIRAC

The HIRAC chamber (Glowacki et al., 2007) is constructed from 304 stainless steel and has an internal volume of ~2.25 m³, the contents of which are homogenised by four mixing fans. Eight 50 mm diameter quartz tubes are mounted radially inside the chamber and extend along its ~2 m length. Each of the eight tubes house a UV lamp that is used to initiate chemical reactions. The lamps can be changed to different wavelength outputs depending on the chemical precursors to be used. The FAGE instrument is connected to the HIRAC chamber through an ISO-K160 flange with an O-ring compression fitting to allow the inlet distance from the wall of the chamber to be varied. The 380 mm long inlet allows the instrument to sample well away from the inner walls of the HIRAC chamber and avoid chemical processes at the metal surface. Because the FAGE system removes gas from the HIRAC chamber, a constant flow of synthetic air is introduced into the chamber to maintain a constant pressure. The CRDS setup is described in Sect. 2.3.

The experiments were conducted inside the HIRAC chamber at 295 K using three different pressure / gas mixtures. The first used 80 mbar total pressure of helium (BOC, >99.99 %) and oxygen (BOC, >99.999 %) in the ratio of $He:O_2 = 3:1$. The second and third mixtures both used synthetic air obtained by mixing oxygen with nitrogen (BOC, > 99.998 %) in the ratio $N_2:O_2 = 4:1$ at 100 and 1000 mbar total pressure, respectively. $CH_3O_2$ was generated in the chamber by photolysing one of two precursor gas mixtures. The first $CH_3O_2$ precursor system was a mixture of $Cl_2$ (Sigma Aldrich, $\geq$ 99.5 %) and $CH_4$ (BOC, CP grade, 99.5 %), where the $Cl_2$ was photolysed at ~365 nm (Phillips, TL-D36W/BLB, $\lambda$ = 350–400 nm) to generate $CH_3O_2$ *via* the reactions:

$$Cl_2 + h\nu \text{ (365 nm)} \rightarrow Cl + Cl \tag{R6}$$

$$CH_4 + Cl \rightarrow CH_3 + HCl \tag{R7}$$

$$CH_3 + O_2 + M \rightarrow CH_3O_2 + M \tag{R1}$$

Typical reagent concentrations were $[CH_4] = 1.2–2.5 \times 10^{16}$ molecule cm⁻³ and $[Cl_2] = 1.1–5.5 \times 10^{15}$ molecule cm⁻³. The second method used the photolysis of acetone (Sigma Aldrich, HPLC grade, $\geq$ 99.9 %) at 254 nm (GE G55T8/OH 7G lamps) to produce $CH_3O_2$ *via* (R9) and (R10) followed by (R1):

$$(CH_3)_2CO + h\nu \text{ (254 nm)} \rightarrow 2CH_3 + CO \tag{R8}$$

$$(CH_3)_2CO + h\nu \text{ (254 nm)} \rightarrow CH_3 + CH_3CO \tag{R9}$$

Typical initial concentrations were $[(CH_3)_2CO] = 8.8 \times 10^{14}$ molecule cm⁻³. In the FAGE calibration experiments using the kinetic decays $[Cl_2]_0 = 1.1 \times 10^{14}$ molecule cm⁻³ with $CH_4$ at one of two concentrations: $2.5 \times 10^{16}$ molecule cm⁻³ and $2.5 \times 10^{17}$ molecule cm⁻³. In the kinetic experiments performed to determine the absorption cross section of $CH_3O_2$ at 7487.98 cm⁻¹, $[Cl_2]_0 = 1.1 \times 10^{14}$ molecule cm⁻³ and $[CH_4]_0 = 2.5 \times 10^{16}$ molecule cm⁻³ at 80 mbar $He:O_2 = 3:1$ and $[Cl_2]_0 = 1.0 \times 10^{15}$ molecule cm⁻³ and $[CH_4]_0 = 2.4 \times 10^{16}$ molecule cm⁻³ at 100 mbar $N_2:O_2 = 4:1$.

## 2.2 FAGE instrument and calibration for $CH_3O_2$

The FAGE instrument in HIRAC has been described in detail previously (Winiberg et al., 2015;Onel et al., 2017a;Onel et al., 2017b). The instrument has a ~ 1 m long black anodised aluminium sampling tube with an inner diameter of 50 mm. The interior of the tube is held at a low pressure (3.3 mbar for a HIRAC pressure, $p_{HIRAC}$ of 1000 mbar of synthetic air and 0.9 mbar for $p_{HIRAC}$ = 100 mbar synthetic air and $p_{HIRAC}$ = 80 mbar mixture of $He:O_2$ = 3:1) and draws sample gas in through a 1 mm diameter pinhole mounted on one end of the tube at a rate of ~3 SLM. Two fluorescence cells are integrated into the tube, the centre of the first cell is ~300 mm from the pinhole, and the centre of the second cell is a further ~300 mm downstream, followed by a line of tubing that is connected to a rotary backed roots blower pump system (Leybold Trivac D40B and Ruvac WAU 251). The first cell is used to detect OH radicals but is not relevant to this work and is not discussed further, whereas the second cell is used for the $CH_3O_2$ measurements detailed here. The $CH_3O_2$ radicals sampled through the FAGE pinhole at 1000 mbar in HIRAC reached the detection region in about 85 ms. High purity NO (BOC, N2.5 nitric oxide) is injected at 2.5 sccm using a mass flow controller (Brooks 5850S) into the centre of the gas flow ~25 mm prior to the second cell to convert $CH_3O_2$ radicals into $CH_3O$. Pulsed laser light at 297.79 nm is directed through the cell and propagates perpendicular to the gas flow and is used to excite the $A^2A_1(v_3' = 3) \leftarrow X^2E(v_3'' = 0)$ transition of $CH_3O$. The off resonant, red shifted fluorescence (320-430 nm) from $CH_3O$ (A) is subsequently detected by a microchannel plate photomultiplier (Photek PMT325) using photon counting. Measurements are made at an excitation wavelength of 297.79 + 2.5 nm in order to determine the laser background, which is subtracted to leave only signal due to $CH_3O$ fluorescence.

The FAGE technique is not absolute and therefore determination of the calibration factor, $C_{CH_3O_2}$ (counts $cm^3$ $molecule^{-1}$ $s^{-1}$ $mW^{-1}$), is required, to convert the measured signal, $S_{CH_3O_2}$ (counts $s^{-1}$ $mW^{-1}$), to the $CH_3O_2$ concentration:

$$[CH_3O_2] = \frac{S_{CH_3O_2}}{C_{CH_3O_2}} \tag{1}$$

### 2.2.1 Calibration at atmospheric pressure - $H_2O$ vapour photolysis in the presence of excess $CH_4$

This calibration procedure has been described in detail previously (Winiberg et al., 2015;Onel et al., 2017b), as such only important points are presented here. $CH_3O_2$ radicals were generated by photolysing water vapour in air (BOC, synthetic BTCA 178) at 184.9 nm to produce OH radicals, which then reacted with methane (BOC, CP grade, 99.5 %) to produce $CH_3O_2$:

$$H_2O + hv \text{ (185 nm)} \rightarrow OH + H \tag{R10}$$

$$OH + CH_4 \rightarrow CH_3 + H_2O \tag{R11}$$

$$CH_3 + O_2 + M \rightarrow CH_3O_2 + M \tag{R1}$$

The subsequent air/radical mixture was then sampled by the FAGE instrument. The concentration of $CH_3O_2$ generated is given by:

$$[CH_3O_2] = [OH] = [H_2O] \cdot \sigma \cdot \Phi \cdot F \cdot t \tag{2}$$

where $\sigma$ is the absorption cross section of water vapour at 184.9 nm, $(7.22 \pm 0.22) \times 10^{-20}$ $cm^2$ $molecule^{-1}$ (Cantrell et al., 1997;Creasey et al., 2000), $\Phi$ is the photodissociation quantum yield of OH at 184.9 nm (unity), $t$ is the residence time of the gas in the photolysis field, which is ~16.6 and ~8.3 ms at 20 and 40 SLM respectively, and $F$ is the lamp flux at 184.9 nm. The product $F \cdot t$ is determined using chemical actinometry (Winiberg et al., 2015). The 184.9 nm photon flux, $F$, is proportional

to the electrical current supplied to the photolysis lamp and is varied to produce a range of $CH_3O_2$ radical concentrations. A typical calibration plot of the FAGE LIF signal vs. generated $[CH_3O_2]$ calculated using Eq. (2) is shown in the Supplementary Information, Figure S2. An average of four calibrations gave $C_{CH_3O_2} = (8.03 \pm 1.37) \times 10^{-10}$ counts $cm^3$ molecule$^{-1}$ s$^{-1}$ mW$^{-1}$ where the error represents the overall uncertainty (17%) calculated using the statistical error (7%) and the systematic error (16%) at $1\sigma$ level (Onel et al., 2017b).

### 2.2.2 Calibration using kinetics of the $CH_3O_2$ temporal decay

The calibration described in the previous section is only valid when FAGE is sampling at atmospheric pressure. However, when sampling from lower pressures, as described in Sect. 2.1, the FAGE cell pressure decreases (0.9 mbar sampling from 100 mbar) and the calibration is no longer valid. An alternative calibration procedure using the kinetics of the $CH_3O_2$ self-reaction inside the HIRAC chamber allowed the FAGE instrument to be calibrated under the same conditions of pressure as the intercomparison experiments, including at lower pressures. Table 1 shows the sensitivity factors, $C_{CH_3O_2}$, obtained for each set of chamber conditions. Radicals were generated in the chamber in the same manner as those described in Sect. 2.1. However, instead of measuring steady state radical concentrations, the lamps were switched off and on at ~120 s intervals to produce a series of second–order decays, typically 4 per experiment, in which $CH_3O_2$ undergoes loss via self-reaction:

$$CH_3O_2 + CH_3O_2 \rightarrow CH_3OH + CH_2O + O_2 \tag{R5a}$$

$$CH_3O_2 + CH_3O_2 \rightarrow CH_3O + CH_3O + O_2 \tag{R5b}$$

Assuming no wall loss for $CH_3O_2$, the kinetic decays can be described by the integrated second order rate equation:

$$\frac{1}{[CH_3O_2]_t} = \frac{1}{[CH_3O_2]_0} + 2 \cdot k_{obs} \cdot \Delta t \tag{3}$$

where $[CH_3O_2]_t$ is the radical concentration at time $t$ of the decay, $[CH_3O_2]_0$ is the initial concentration at the time $t_0$, when the lamps are switched off, $\Delta t = t - t_0$ and $k_{obs}$ is the observed rate coefficient. The observed rate coefficient is larger than the second order rate coefficient of just the $CH_3O_2$ recombination reaction (R5) as the methoxy radicals generated by channel R5.b react rapidly with oxygen present in large excess to produce $HO_2$ (R3), which in turn reacts with $CH_3O_2$ (R12).

$$CH_3O + O_2 \rightarrow CH_2O + HO_2 \tag{R3}$$

$$CH_3O_2 + HO_2 \rightarrow 0.9CH_3OOH + 0.1CH_2O + 0.1H_2O + O_2 \tag{R12}$$

As each $HO_2$ radical consumes rapidly one $CH_3O_2$ species on the time scale of the reaction R5, the $CH_3O_2$ decay is described by second order kinetics (Sander and Watson, 1980); (Sander and Watson, 1981;McAdam et al., 1987;Kurylo and Wallington, 1987;Jenkin et al., 1988;Simon et al., 1990), with $k_{obs} = k_5(1 + r_{5b})$, where $r_{5b}$ is the branching ratio for the channel R5b. By using the IUPAC recommendations (Atkinson et al., 2006): $k_5 = (3.5 \pm 1.0) \times 10^{-13}$ molecule$^{-1}$ cm$^3$ s$^{-1}$ and $r_{5b} = 0.37 \pm 0.06$, a value of $4.8 \times 10^{-13}$ molecule$^{-1}$ cm$^3$ s$^{-1}$ is obtained for $k_{obs}$.

Modelling of the decay process with a variety of $CH_3O_2$ and $HO_2$ concentrations after the lamps were switched off and following the establishment of steady state conditions showed that Eq. (3) was valid within experimental error. With $k_5 = 3.5 \times 10^{-13}$ molecule$^{-1}$ cm$^3$ s$^{-1}$ (Atkinson et al., 2006), a faster observed rate constant (defined by Eq. (3)) was obtained from the model with a value, $4.9 \times 10^{-13}$ molecule$^{-1}$ cm$^3$ s$^{-1}$ consistent with that recommended by IUPAC, $(4.8 \pm 0.6) \times 10^{-13}$ molecule$^{-1}$

cm$^3$ s$^{-1}$ (1$\sigma$ uncertainty; Atkinson et al., 2006). Substituting Eq. (1) into Eq. (3) allows the measured signal over the decay to be related to the instrument sensitivity by:

$$\frac{1}{(S_{CH_3O_2})_t} = \frac{1}{(S_{CH_3O_2})_0} + \frac{2 \cdot k_{obs} \cdot \Delta t}{C_{CH_3O_2}} \tag{4}$$

where $(S_{CH_3O_2})_t$ and $(S_{CH_3O_2})_0$ are the FAGE signal at time $t$ and $t_0$ respectively. Taking the reciprocal of Eq. (4) gives:

$$(S_{CH_3O_2})_t = \left( \frac{1}{(S_{CH_3O_2})_0} + \frac{2 \cdot k_{obs} \cdot \Delta t}{C_{CH_3O_2}} \right)^{-1} \tag{5}$$

which is then used to fit to the experimental data with $k_{obs}$ fixed to the value recommended by IUPAC for 298 K, $4.8 \times 10^{-13}$ molecule$^{-1}$ cm$^3$ s$^{-1}$, using the Levenberg-Marquardt algorithm. Figure 1 shows an example CH$_3$O$_2$ self-reaction decay trace

10    obtained at 1000 mbar, where the red line shows the result of the fitting process.

However, as the HIRAC chamber is constructed from steel, the potential for a loss of CH$_3$O$_2$ to the walls was investigated. As circulation fans were used during all the experiments, the 'movement' of CH$_3$O$_2$ radicals within the chamber is in part molecular diffusion and in part convection. Therefore, the parameter $k_{loss}$ is controlled by both convection and diffusion processes. By incorporating the wall loss as a first-order process Eq. (5) becomes:

$$(S_{CH_3O_2})_t = \left( \left( \frac{1}{(S_{CH_3O_2})_0} + \frac{2 \cdot k_{obs}}{k_{loss} \cdot C_{CH_3O_2}} \right) \times \exp(k_{loss} \cdot \Delta t) - \left( \frac{2 \cdot k_{obs}}{k_{loss} \cdot C_{CH_3O_2}} \right) \right)^{-1} \tag{6}$$

Fitting Eqs. (5) and (6) to the experimental data is also shown in Fig. 1. The extracted values for the sensitivity factor are the same for the fit without and with wall loss included: $C_{CH3O2} = (1.17 \pm 0.04) \times 10^{-9}$ counts cm$^3$ molecule$^{-1}$ s$^{-1}$ mW$^{-1}$ (statistical

20    errors at 1$\sigma$ level). The close overlap of the fits without and with wall loss included and the small values extracted for $k_{loss}$ (upper limit of $\sim 1 \times 10^{-5}$ s$^{-1}$) fitting Eq. (6) demonstrates that wall losses are very small and can be neglected. This is evidenced further by the lack of an observable radical gradient across the chamber diameter as shown in Fig. S5 in the Supplementary Information. In addition, modelling the CH$_3$O$_2$ decays including a wall loss for HO$_2$ in the range of measured values 0.03–0.09 s$^{-1}$ (Onel et al. 2017a), showed a minor impact of the wall loss of HO$_2$ on $k_{obs}$, i.e. $k_{obs}$ within 98–95 % agreement with

25    the IUPAC preferred value, $(4.8 \pm 0.6) \times 10^{-13}$ molecule$^{-1}$ cm$^3$ s$^{-1}$ (1$\sigma$ uncertainty; Atkinson et al., 2006).

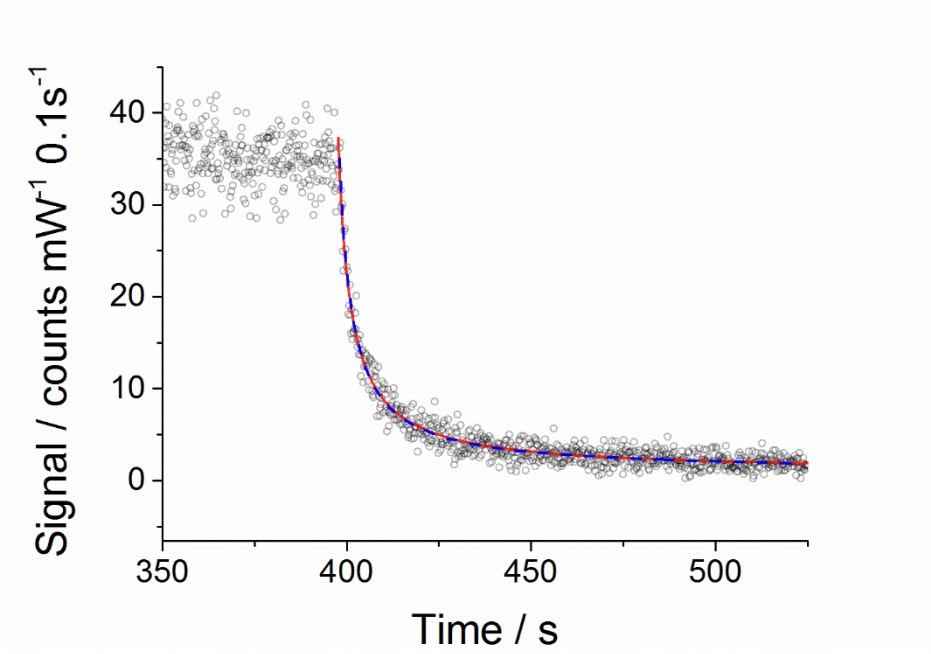

**Figure 1.** An example of a second order decay of the FAGE $CH_3O_2$ signal (normalized for laser power fluctuations) with 0.1 second time resolution (black open circles) recorded at 295 K and a 1000 mbar air mixture. $CH_3O_2$ was generated using $[Cl_2] \sim 1.1 \times 10^{14}$ molecule $cm^{-1}$ and $[CH_4] \sim 2.5 \times 10^{16}$ molecule $cm^{-3}$. At time zero (~400 s) the photolysis lamps were turned off to allow the radicals to decay. The data were fitted to Eq. (5) (excluding the wall loss rate, $k_{loss}$; red line) and Eq. (6) (including $k_{loss}$; blue dashed line) using the Levenberg-Marquardt algorithm. The obtained value for the sensitivity factor was the same for both fits: $C_{CH3O2} = (1.17 \pm 0.04) \times 10^{-9}$ counts $cm^3$ molecule$^{-1}$ s$^{-1}$ mW$^{-1}$. The $C_{CH3O2}$ errors given above represent statistical uncertainties at $1\sigma$ level.

**Table 1.** Average sensitivity factors for the FAGE instrument using the $CH_3O_2$ kinetic decay method under each chamber environment. Examples of these decays can be found in Figure 1 above and in the SI, Figures S3 and S4, and reported values are typically from an average of 8 decays. All the data were fitted using Eq. (5). The errors given in the table are overall uncertainties (13%) at $1\sigma$ level.

| Chamber Conditions | $C_{CH_3O_2}$ / counts cm$^3$ molecule$^{-1}$ s$^{-1}$ mW$^{-1}$ |
|---|---|
| 80 mbar, He + $O_2$ | $(3.83 \pm 0.50) \times 10^{-9}$ |
| 100 mbar, Air | $(2.80 \pm 0.37) \times 10^{-9}$ |
| 1000 mbar, Air | $(1.16 \pm 0.15) \times 10^{-9}$ |

Table 1 shows the average sensitivity factors obtained fitting Eq. (5) to a typical number of 8 temporal decays of $S_{CH_3O_2}$ under each of the chamber conditions, and example decay traces for the 80 and 100 mbar experiments can be found in the SI, Figs S3 and S4, respectively. These factors are used for their respective experimental conditions. For the 1000 mbar intercomparison experiments with CRDS, an average of the water vapour photolysis sensitivity factor at 1000 mbar, $C_{CH3O2, H2O} = (8.03 \pm 1.37) \times 10^{-10}$ counts $cm^3$ molecule$^{-1}$ s$^{-1}$ mW$^{-1}$, and the average sensitivity factor obtained from the kinetic decay, $C_{CH3O2, kinetic} = (1.16 \pm 0.15) \times 10^{-9}$ counts $cm^3$ molecule$^{-1}$ s$^{-1}$ mW$^{-1}$ (Table 1), is used, giving $C_{CH3O2, av.} = (9.81 \pm 2.03) \times 10^{-10}$ counts $cm^3$ molecule$^{-1}$ s$^{-1}$ mW$^{-1}$. We make a brief comment regarding the difference in the sensitivity factors $C_{CH3O2, H2O}$ and $C_{CH3O2, kinetic}$, for which

the ratio is ~ 0.7, showing a ~30% difference, although the two calibration methods have overlapping error limits at $2\sigma$ level. The kinetic method relies on the rate coefficient $k_{obs}$ for the $CH_3O_2$ self-reaction as recommended by IUPAC (Atkinson et al., 2006), which has a quoted $2\sigma$ uncertainty of 23%. In a separate paper we will present a detailed study of the kinetics of the $CH_3O_2$ self-reaction, and its temperature dependence, and report a revised rate coefficient for this reaction at 298 K.

As the pressure in the FAGE detection cell was 2-3 orders of magnitude lower than the corresponding pressure in HIRAC (*vide supra* in Sect. 2.2) the concentrations of the reagents ($Cl_2$, methane and acetone) were also 2-3 orders of magnitude lower in the fluorescence cells than the reagent concentrations in HIRAC. However, a potential effect of the reagents ($Cl_2$, methane and acetone) on the FAGE sensitivity factor in the HIRAC experiments was investigated. Two different concentrations of $CH_4$ were used in the kinetic method for FAGE calibration at 80 mbar of He + $O_2$ in HIRAC to find practically the same sensitivity

factor: $(3.80 \pm 0.50) \times 10^{-9}$ counts $cm^3$ molecule$^{-1}$ s$^{-1}$ mW$^{-1}$ for $2.5 \times 10^{16}$ molecule $cm^{-3}$ $CH_4$ ($2.8 \times 10^{14}$ molecule $cm^{-3}$ in the fluorescence cell) and $(3.86 \pm 0.50) \times 10^{-9}$ counts $cm^3$ molecule$^{-1}$ s$^{-1}$ mW$^{-1}$ for $2.5 \times 10^{17}$ molecule $cm^{-3}$ $CH_4$ ($2.8 \times 10^{15}$ molecule $cm^{-3}$ in the fluorescence cell). As shown in Fig. S1 in the Supplement there is a good agreement between the laser excitation scans of $CH_3O$ obtained from the $CH_3O_2$ generated in HIRAC using the two methods: acetone photolysis and $Cl_2$ photolysis in the presence of $CH_4$ and $O_2$. In addition, a good agreement has been previously found between the laser excitation

spectra of $CH_3O$ generated using the reaction of $CH_4$ with OH (generated by the 254 nm photolysis of water) in the presence of $O_2$ and directly, through the 254 nm photolysis of $CH_3OH$. Therefore, no effect of the used reagents on the laser excitation spectrum of $CH_3O$ was found.

**2.2.3 FAGE measurements of $CH_3O_2$ concentration gradient across the HIRAC diameter**

Measurement of radical gradients across the chamber diameter have been performed previously for $HO_2$ radicals (Onel et al., 2017a), where no gradient was observed until measuring <10 cm from the chamber wall where the signal began to decrease, ultimately by ~16 % at the point at which the FAGE sampling pinhole was level with the chamber walls. To investigate any similar gradient effects for $CH_3O_2$, a steady state concentration of $CH_3O_2$ was generated in the chamber at atmospheric pressure

by photolysing $O_3$ in the presence of air and methane:

$$O_3 + h\nu \text{ (254 nm)} \rightarrow O_2 + O(^1D) \tag{R12}$$
$$O(^1D) + CH_4 \rightarrow CH_3 + OH \tag{R13}$$
$$CH_3 + O_2 + M \rightarrow CH_3O_2 + M \tag{R1}$$


Ozone and methane were present in the chamber at ~$2.5 \times 10^{13}$ molecule $cm^{-3}$ and $2.5 \times 10^{17}$ molecule $cm^{-3}$ respectively. The FAGE inlet was translated across the width of the chamber and the $CH_3O_2$ signal was observed to show no decrease within the ~10% $1\sigma$ statistical error of each measurement up until the point at which the pinhole was level with the chamber walls. Moving the instrument further backwards positioned the pinhole inside the ISO-K160 coupling flange and effectively ~4 cm

behind the chamber walls where there is likely to be little air movement. This position is analogous to that of the CRDS mirrors, which are recessed into the chamber walls as they mount to the outside of the chamber (see Sect. 2.3). In this position a signal drop of ~14 % was observed, within the statistical error margins of the measurements. A plot of the radical gradient is shown in the Supplementary Information, Figure S5.

**2.3 CRDS set-up**

The optical path of the CRDS spectrometer within the HIRAC chamber is shown in Fig. 2 and is the same spectrometer as used to probe $HO_2$ across the chamber's diameter, and which has been described previously (Onel et al., 2017a).

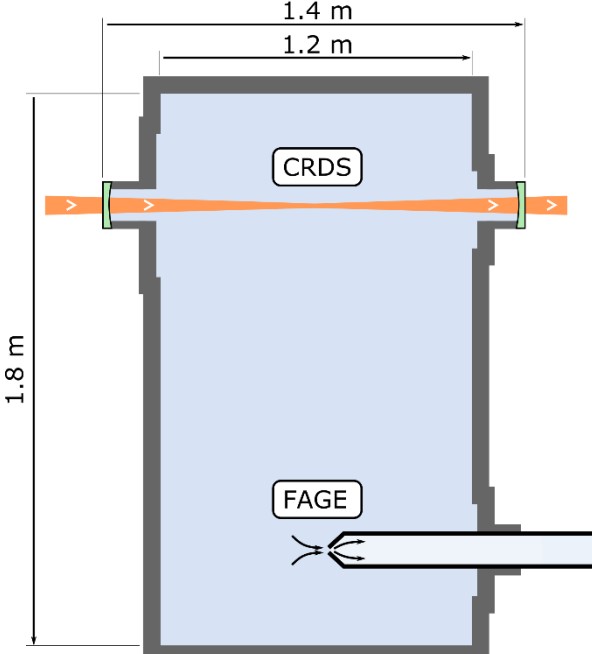

**Figure 2.** Longitudinal (horizontal) section of the HIRAC chamber. The CRDS spectrometer probes the $CH_3O_2$ concentration as an average across the chamber's diameter, while the FAGE instrument probes $CH_3O_2$ in the chamber at a single point close to the centre.

The cavity is formed by two highly reflective 1 in. diameter mirrors (99.999 %, Layertec, curvature radius = 1 m) housed in custom built mounts that allow the mirrors to be tilted slightly whilst maintaining a gas-tight seal. The position of the mirror on the laser injection side is modulated along the cavity axis by a few microns using a piezoelectric transducer at ~10 Hz, with the overall distance between the two mirrors being ~1.4 m. Laser light of ~1.335 μm is generated by a distributed feedback (DFB) fibre pig-tailed diode laser (NTT Electronics, NLK1B5EAAA) held in a butterfly laser diode mount (Thorlabs LM14S2). The electrical current that drives the laser diode and thermoelectric cooler is generated by a Thorlabs ITC502 driver. The DFB is connected to an inline optical isolator (Thorlabs IO-H-1335APC), an acousto-optic modulator (AOM, Gooch & Housego Fibre-Q M040-0.5C8H-3-F2S) and a fibre collimator (Thorlabs CFC-8X-C). The laser light is then guided into the cavity by two silver mirrors (Thorlabs PF10-03-P01). On the detection side of the cavity, light leaking out of the mirror is directed onto another silver mirror that guides the light through a $f$ = 30 mm focusing lens (Thorlabs LA1805-C) onto an InGaAs photodiode (Thorlabs DET10C/M) that is isolated from ambient light by a 1250 nm longpass filter (Thorlabs FELH1250). The photodiode signal is amplified (FEMTO DLPCA-200) and sent to a data acquisition unit (DAQ, National Instruments USB-6361) and to a custom-built comparator that acts as a trigger unit. The comparator compares the amplified photodiode signal with a manually adjustable threshold voltage, and upon reaching a preset threshold the AOM is switched off, stopping the injection of light into the cavity within tens of nanoseconds and initiating a ring-down event. The DAQ is simultaneously triggered and acquires the signal ring-down. The system resets after a set time (typically 5 ms) ready for the next event. The acquired data are processed using a custom made LabView program that fits the ring-down events with an exponential function to extract the ring-down time, $\tau$. Filters are applied to process the ring-down events to exclude potential outliers caused by dust particles passing through the beam and false positives (when the acquisition is triggered by a transient

noise spike), so that only legitimate ring-down events are taken into account. The ring-down time can then be converted into the absorption coefficient, $\alpha$:

$$\alpha = \frac{1}{c} \times (1/\tau - 1/\tau_0) \tag{7}$$

where $\tau$ and $\tau_0$ are the ring-down times with and without $CH_3O_2$ radicals present, respectively, and $c$ is the speed of light. $\tau_0$ would be obtained in a typical experiment by recording ring-down events for ~ 1 minute before switching on the photolysis lamps in the chamber. As it is not possible to measure $\tau_0$ and $\tau$ simultaneously, the background was monitored regularly during each experiment by switching off the photolysis lamps and allowing the signal to return to the baseline.

The molecular chlorine delivery did not result in a change in the measured ring-down time. However, delivery of the methane and acetone reagents led to a decrease in the ring-down time indicating that, in the concentrations delivered to the chamber, methane and acetone absorbed in the wavenumber range used in the present work, ~7486–7491 cm$^{-1}$. An absorption coefficient of ~$8 \times 10^{-9}$ cm$^{-1}$ was measured for [acetone] $\approx 9 \times 10^{14}$ molecule cm$^{-3}$ at the typical measurement point of 7487.98 cm$^{-1}$ (*vide infra*). An absorption coefficient in the range $(0.7–1.4) \times 10^{-8}$ cm$^{-1}$ was determined at 7487.98 cm$^{-1}$ for $CH_4$ in typical
concentrations in the FAGE–CRDS intercomparison experiments in the range $(1.2–2.5) \times 10^{16}$ molecule cm$^{-3}$. The background ring-down time $\tau_0$ (Eq. 7) contained the contributions of the reagents, methane or acetone, and was monitored regularly during the experiments by turning off the chamber lamps (*vide supra*).

The $CH_3O_2$ absorption feature used in these measurements is a band associated with the $A^2A' \leftarrow X^2A''$ electronic transition centred around 7488 cm$^{-1}$, and has been documented in previous work (Faragó et al., 2013, Atkinson and Spillman, 2002,
Pushkarsky et al., 2000). There are interfering methane and water vapour lines in this region, and these together with the change in [$CH_3O_2$] during longer ($\gtrsim$ 10 min) scanning times did not allow us to generate a continuous, high resolution scan across the $CH_3O_2$ transition. Instead, as shown in Fig. 3, the absorption spectrum was mapped out as a series of point measurements at fixed wavelengths, normalised by the absorption at the optimum measurement point, 7487.98 cm$^{-1}$, where the absorption feature is sufficiently strong and furthest in wavelength from interfering methane absorption lines and where
the $CH_3O_2$ cross section was determined (Sect. 3.2). The absorption coefficient of $CH_4$ was about 7 times lower at 7487.98 cm$^{-1}$ than at 7489.16 cm$^{-1}$, i.e. at the peak of the $CH_3O_2$ spectral feature where Fittschen (2019) reported $\sigma_{CH3O2}$. Therefore, 7487.98 cm$^{-1}$ (rounded to 7488 cm$^{-1}$ henceforth) was chosen as the measurement point instead of the value of 7489.16 cm$^{-1}$ used by Fittschen (2019). Each datum point in Fig. 3 was obtained by measuring the absorption coefficient, $\alpha_{7488\ cm^{-1}}$, and the baseline (lamps on, then off) at 7488 cm$^{-1}$ followed by measuring $\alpha_{CH3O2}$ and baseline at another wavelength on the absorption
feature and then reverting to measuring at 7488 cm$^{-1}$ again. This pattern was repeated multiple times for different wavelengths to build up an absorption feature, with all data points normalised to $\alpha_{7488\ cm^{-1}}$ and then multiplied by the $CH_3O_2$ cross section at 7488 cm$^{-1}$ (Sect. 3.2) to obtain the absorption spectrum shown in Fig. 3. The method was used to measure the $CH_3O_2$ absorption spectrum under each of the three experimental conditions detailed in Sect. 2.1: 80 mbar (He + $O_2$) and 100 mbar and 1000 mbar of synthetic air.


## 3 Results

### 3.1 CH₃O₂ absorption spectrum and comparison with the literature

Figure 3 shows that the relatively broad absorption feature obtained in this work in the range from ~7486 to 7491 cm$^{-1}$ is
almost the same at 80 mbar He:$O_2$ = 3:1 and at 100 and 1000 mbar of synthetic air. As shown in Fig. 3, the spectrum found in this work agrees well with the general shape of the $CH_3O_2$ spectrum measured by Faragó et al. (2013) at 67 mbar He:$O_2$ ~ 1:1 but scaled to reflect the very recent update to the absolute absorption cross-section reported by Fittschen (2019) which gave

$\sigma_{7489\ cm^{-1}} = 2.2 \times 10^{-20}$ cm$^2$ molecule$^{-1}$. The peaks at the top of the spectral feature reported by Faragó et al. (2013) are not reproduced in this work owing to the method of generating the spectrum (Sect. 2.3). Previously Pushkarsky et al. (2000) measured the CH$_3$O$_2$ absorption spectrum over a larger wavenumber range (7300–7700 cm$^{-1}$) where the $\nu_{12}$ transition is located at 7488 cm$^{-1}$ in agreement with this work. In addition, if the CH$_3$O$_2$ spectrum at 27 mbar N$_2$:O$_2$ = 4:1 reported by Atkinson and

Spillman (2002) were shifted by ~2 cm$^{-1}$ toward higher wavenumbers compared to this work and the study of Faragó et al. (2013), the shape of the $\nu_{12}$ band from Atkinson and Spillman is in agreement with the results shown in Fig. 3.

The similarity of the results at 80 mbar He:O$_2$ = 3:1 and at 100 and 1000 mbar of air reported in this work and their agreement with the previous measurements performed at relatively low pressures (Fittschen, 2019; Faragó et al., 2013; Atkinson and Spillman, 2002; Pushkarsky et al., 2000) can be explained by an overlap of several individual absorption lines

resulting in a spectral structure in the range from ~7486 to 7491 cm$^{-1}$ with practically no pressure dependence observed between ~30–1000 mbar. Therefore, it can be assumed that the absorption cross section at 7488 cm$^{-1}$, $\sigma$(7488 cm$^{-1}$), is the same under the conditions used in this work, i.e. at 80 mbar of He and O$_2$ and at 100 and 1000 mbar of air.

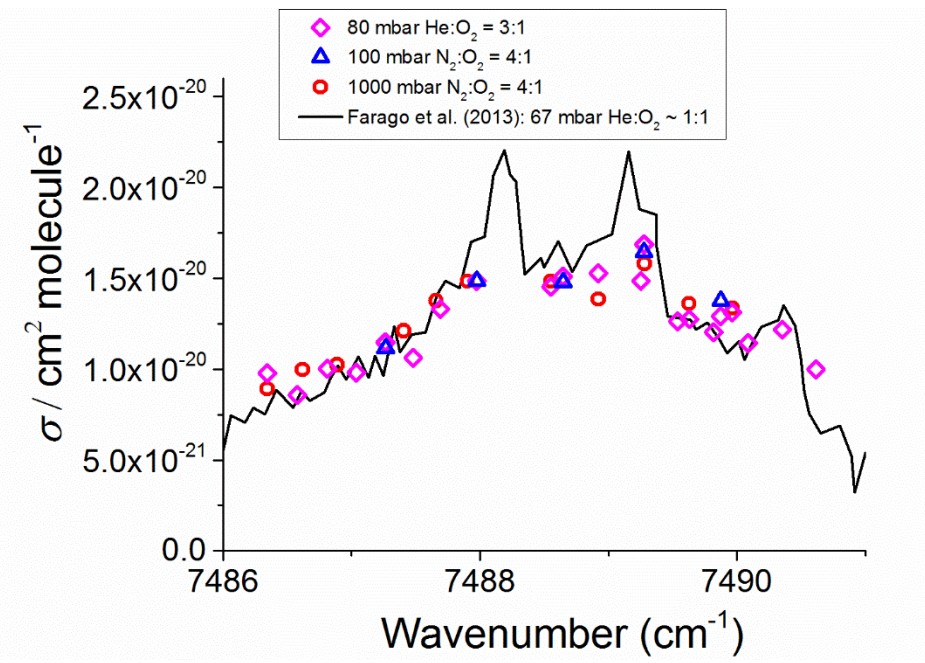

**Figure 3**. CH$_3$O$_2$ absorption spectrum at 295 K. The measured absorption spectrum scaled to the absolute cross section determined at 7488 cm$^{-1}$ using the kinetics of the CH$_3$O$_2$ decay monitored using CRDS (Sect. 3.2 below). The black line represents the CH$_3$O$_2$ spectrum measured by Faragó et al. (2013) at 67 mbar He:O$_2$ ~ 1:1 but with the absolute cross-section

scaled to reflect the recent update reported by Fittschen (2019) giving $\sigma_{7489\ cm^{-1}} = 2.2 \times 10^{-20}$ cm$^2$ molecule$^{-1}$.

**3.2 Determination of the absorption cross section of CH$_3$O$_2$ at 7488 cm$^{-1}$**

The kinetics of the CH$_3$O$_2$ temporal decay by its self-reaction (Reaction R5) were used to determine the absorption cross section of CH$_3$O$_2$ at 7488 cm$^{-1}$, $\sigma$(7488 cm$^{-1}$). Note that the cross-section used is not the more standard integrated cross-section

used by HITRAN and other spectral databases. CH$_3$O$_2$ radicals were generated by using CH$_4$/Cl$_2$/synthetic air mixtures (Sect. 2.1) with the chamber UV lamps switched on to generate Cl atoms (R6). By extinguishing the UV lamps, CH$_3$O$_2$ radicals were removed by self-reaction and wall loss. Figure 4 shows an example of a kinetic decay obtained at 100 mbar N$_2$:O$_2$ = 4:1 using CRDS. The experimental data were fitted by using two functions described by Eqs. (8) and (9), which are closely related to Eqs. (4) and (5) used in the analysis of the CH$_3$O$_2$ decays measured using FAGE. Equation (8) assumes that the wall loss of

CH$_3$O$_2$ is negligible and hence the removal of CH$_3$O$_2$ can be described by the integrated second–order rate law equation, leading to:

$$\alpha_t = \left(\frac{1}{\alpha_0} + \frac{2 \cdot k_{obs.}\Delta t}{\sigma(7488\ cm^{-1})}\right)^{-1}, \tag{8}$$

where $\alpha_t$ is the $CH_3O_2$ absorption coefficient at 7488 cm$^{-1}$ and at time $t$, $\alpha_0$ is the absorption coefficient at time zero of the reaction when the lamps are switched off, $t_0$, $\Delta t = t - t_0$ and $k_{obs}$ is the observed rate coefficient of the self-reaction at 298 K, $k_{obs} = (4.8 \pm 0.6) \times 10^{-13}$ cm$^3$ molecule$^{-1}$ s$^{-1}$ (Atkinson et al., 2006).

For completeness, Equation (9) includes the $CH_3O_2$ wall loss as a first-order process, leading to:

$$\alpha_t = \left(\left(\frac{1}{\alpha_0} + \frac{2 \cdot k_{obs.}}{k_{loss} \cdot \sigma(7488\ cm^{-1})}\right) \times \exp(k_{loss}\Delta t) - \left(\frac{2 \cdot k_{obs.}}{k_{loss} \cdot \sigma(7488\ cm^{-1})}\right)\right)^{-1}, \tag{9}$$

where $k_{loss}$ is the rate coefficient describing the $CH_3O_2$ wall loss (Onel et al., 2017a).

Figure 4 shows that the fits given by Eqs. (8) and (9) to the data overlap over all of the temporal $CH_3O_2$ decay and the values of $\sigma(7488\ cm^{-1})$ extracted by the two fits are in a very good agreement: $(1.47 \pm 0.07) \times 10^{-20}$ cm$^2$ molecule$^{-1}$ (Eq. (8)) and $(1.50 \pm 0.07) \times 10^{-20}$ cm$^2$ molecule$^{-1}$ (Eq. (9)), where the quoted errors are statistical uncertainties. The values extracted for $k_{loss}$ by fitting Eq. (9) to the CRDS data were small and similar to the values obtained by fitting Eq. (6) to the kinetic decays monitored by FAGE. An upper limit of $\sim 1 \times 10^{-5}$ s$^{-1}$ was obtained for $k_{loss}$ in both FAGE and CRDS measurements, showing that wall losses are negligible. From fitting Eq. (8) to the temporal decays obtained at 100 mbar of synthetic air, an averaged value of $(1.51 \pm 0.19) \times 10^{-20}$ cm$^2$ molecule$^{-1}$ was obtained, where the error represents 1$\sigma$ overall uncertainty (13%). Fitting Eq. (8) to the data at 80 mbar He:O$_2$ = 3:1 (Fig. S6), gave an average value of $\sigma(7488\ cm^{-1}) = (1.46 \pm 0.17) \times 10^{-20}$ cm$^2$ molecule$^{-1}$ (1$\sigma$ overall uncertainty), in very good agreement with the value at 100 mbar of air. The average of the results at 80 mbar He:O$_2$ = 3:1 and 100 mbar of air, $1.49 \times 10^{-20}$ cm$^2$ molecule$^{-1}$, is in excellent agreement with the determination of Atkinson and Spillman (2002): $\sigma_{max}(\nu_{12}) = (1.5 \pm 0.8) \times 10^{-20}$ cm$^2$ molecule$^{-1}$ and consistent with the estimation of $\sim 1.0 \times 10^{-20}$ cm$^2$ molecule$^{-1}$ for $\sigma_{max}(\nu_{12})$ obtained using the $CH_3O_2$ spectrum reported by Pushkarsky et al. (2000). To enable a comparison at 7487.98 cm$^{-1}$ with the very recent measurement of Fittschen (2019), who found $2.20 \times 10^{-20}$ cm$^2$ molecule$^{-1}$ at 7489.16 cm$^{-1}$, $\sigma(7487.98\ cm^{-1}) = 1.49 \times 10^{-20}$ cm$^2$ molecule$^{-1}$ obtained in this work was multiplied by the $\sigma(7489.16\ cm^{-1}):\sigma(7487.98\ cm^{-1})$ ratio obtained by using the high resolution spectrum reported by Faragó et al. (2013) (Fig. 3). The obtained value, $\sigma(7489.16\ cm^{-1}) = (1.9 \pm 0.3) \times 10^{-20}$ cm$^2$ molecule$^{-1}$ is in reasonable agreement with the result of Fittschen (2019), $\sigma(7489.16\ cm^{-1}) = 2.2 \times 10^{-20}$ cm$^2$ molecule$^{-1}$.

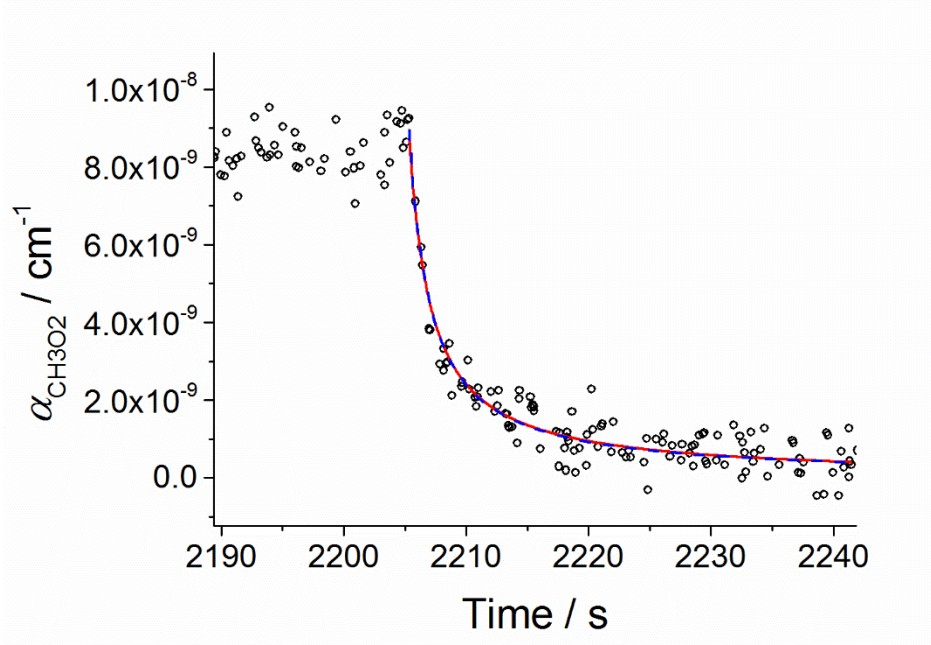

**Figure 4.** Second order decay of the $CH_3O_2$ absorption coefficient at 7488 cm$^{-1}$ monitored by CRDS. Experiment carried out at 295 K and 100 mbar of synthetic air; $[CH_4]_0 = 2.4 \times 10^{16}$ molecule cm$^{-3}$ and $[Cl_2]_0 = 1.0 \times 10^{15}$ molecule cm$^{-3}$. At time 2205 s the photolysis lamps were turned off (time $t_0$). Fitting Eq. (8) to the data (red line) gave $\sigma(7488$ cm$^{-1}) = (1.47 \pm 0.07) \times 10^{-20}$ cm$^2$ molecule$^{-1}$. A fit including the wall loss rate, $k_{loss}$ (Eq. (9)) is shown by the blue dashed line and resulted in $\sigma(7488$ cm$^{-1})$ $= (1.50 \pm 0.07) \times 10^{-20}$ cm$^2$ molecule$^{-1}$. The error limits are statistical errors at $1\sigma$ level.

### 3.3 Determination of the CRDS limit of detection

The CRDS limit of detection (*LOD*) has been computed using plots of the square root of the Allan-Werle variance (Werle et al., 1993;Onel et al., 2017a) obtained by continuously recording single ring-down events for 1–2 hr after delivering either acetone or methane in typical concentrations to the chamber filled with the bath gas (He:$O_2$ = 3:1 at 80 mbar and synthetic air at 100 and 1000 mbar, respectively). The square root of the Allan-Werle variance, here referred to as the Allan-Werle deviation, $\sigma_A(n)$, gives an estimate of the error, $\delta\alpha$, between successively measured absorption coefficients for a given averaging size $n$.

For a signal-to-noise ratio (*S/N*) of 2, the limit of detection for $CH_3O_2$ was determined as $LOD_{CH3O2} = (2\delta\alpha_{min})/\sigma_{CH3O2}$, where $\sigma_{CH3O2} = 1.49 \times 10^{-20}$ cm$^2$ molecule$^{-1}$ is the $CH_3O_2$ cross section at 7488 cm$^{-1}$, and is shown in Table 2. The optimum CRDS sensitivity under all conditions is achieved averaging ~500 ring-down events, requiring ~77 s at an acquisition rate of 6.5 Hz on average, with an example shown in Fig. 5.

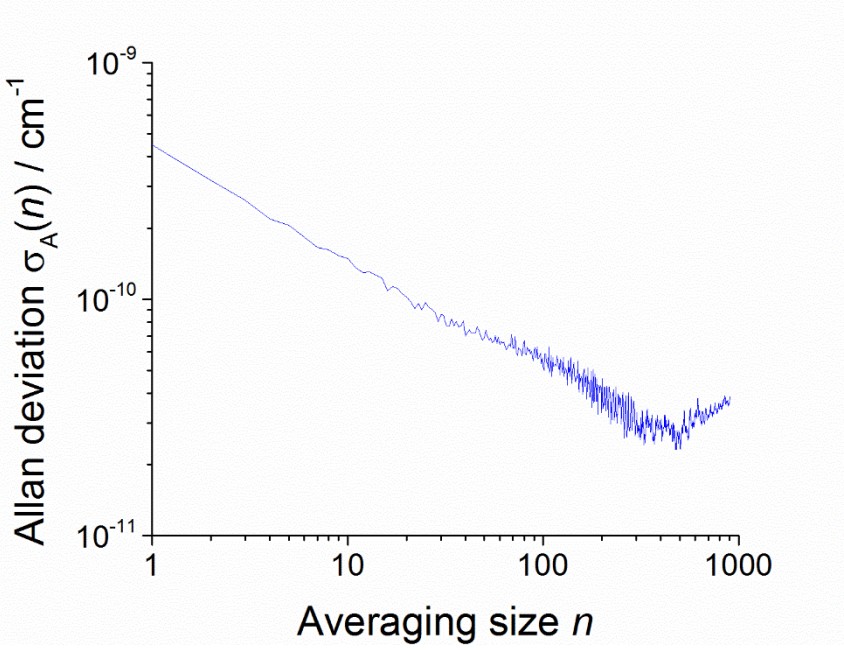

**Figure 5.** An example of the Allan-Werle deviation plot (the plot of the square root of the Allan-Werle variance) of the absorption coefficient at 7488 cm$^{-1}$ in the absence of $CH_3O_2$ and the presence of a typical acetone concentration of $8.8 \times 10^{14}$ molecule cm$^{-3}$ at 1000 mbar against the number of ring-down events averaged, $n$. For $S/N = 2$ the minimum detectable absorption coefficient for a single ring-down measurement is $4.5 \times 10^{-10}$ cm$^{-1}$, which decreases to a minimum of $2.89 \times 10^{-11}$ cm$^{-1}$ after $n = 500$ (requiring 77 s at an acquisition rate of 6.5 Hz).

As the filter (FELH1250 Thorlabs, cut-off wavelength: 1250 nm) used to cut-off the laboratory visible light from the background of the CRDS measurements allowed some of the 254 nm light generated by the HIRAC lamps to be transmitted and then detected by the InGaAS photodiode detector, the CRDS sensitivity was worse in the experiments using acetone/$O_2$/254 nm lamps as a source of $CH_3O_2$ compared to the experiments using $Cl_2$/$CH_4$/$O_2$/UV black lamps to generate $CH_3O_2$. Therefore, separate Allan-Werle deviation plots were constructed using measurements of single ring-down events after filling HIRAC with the bath gas and turning the 254 nm lamps on. Then, the composite error, calculated as the sum in quadrature of $\delta\alpha$ obtained in the presence of acetone and $\delta\alpha$ determined in the absence of acetone but keeping the 254 nm lamps turned on, was used to determine the *LOD* of CRDS in the acetone/$O_2$/254 nm experiments (Table 2). The composite *LOD*(acetone/$O_2$/254 nm lamps) was on average ~55% greater than the *LOD* determined with the UV lamps off and in the absence of acetone, *LOD*(bath gas); and on average *LOD*($Cl_2$/$CH_4$/$O_2$/UV black lamps) was ~40% higher than *LOD*(bath gas)

**Table 2**. CRDS detection limits for $CH_3O_2$ calculated at 80 mbar He:$O_2$ = 3:1 and 100 mbar and 1000 mbar of synthetic air for single ring-down measurements ($\Delta t = 0.15$ s), the optimum averaging time obtained from Allan-Werle deviation plot (Fig. 5 shows an example), $\Delta t_{opt.}$ (77 s under all experimental conditions) and $\Delta t = 60$ s.

| Bath gas | $p_{HIRAC}$ / mbar | Reagent delivered to HIRAC | $LOD_{CH3O2}$ / $10^9$ molecule cm$^{-3}$ | | |
|---|---|---|---|---|---|
| | | | $\Delta t = 0.15$ s | $\Delta t = 60$ s | $\Delta t_{opt} = 77$ s. |
| He:$O_2$ = 3:1 | 80 | acetone[a] | 120 | 7.5 | 6.4 |
| Air | 100 | acetone[a] | 133 | 8.6 | 6.8 |
| | | methane[b] | 78 | 6.0 | 5.4 |
| Air | 1000 | acetone[a] | 147 | 7.3 | 6.1 |

[a] using the composite error calculated as the sum in quadrature of $\delta\alpha$ obtained using a typical concentration of acetone, $8.8 \times 10^{14}$ molecule cm$^{-3}$, and $\delta\alpha$ determined in the absence of acetone but keeping the 254 nm lamps turned on during all measurement.
[b] [$CH_4$] = $2.4 \times 10^{16}$ molecule cm$^{-3}$.

As the daytime concentrations of $CH_3O_2$ have been calculated using an atmospheric box-model to peak at $\sim 10^7 - 10^8$ molecule $cm^{-3}$ (Whalley et al., 2010;Whalley et al., 2011;Whalley et al., 2018), the current CRDS sensitivity is insufficient for

the detection of ambient $[CH_3O_2]$. The typical concentrations of $CH_4$ and acetone in ambient air are orders of magnitude lower than $[CH_4]$ and $[(CH_3)_2CO]$ used in the HIRAC experiments. However, water vapour, which is present in the atmosphere in much larger concentrations (typically $\sim 10^{17}$ molecule $cm^{-3}$) than in HIRAC for these experiments ($\sim 10^{13} - 10^{14}$ molecule $cm^{-3}$), will significantly absorb in this wavelength region and contribute towards the background of the measurements. The limits of detection shown in Table 2 allow for HIRAC measurements of $[CH_3O_2] \gtrsim 10^{10}$ molecule $cm^{-3}$ in steady-state (where averaging

times of $\sim 60$ s can be used) under all conditions used, and kinetic measurements of $[CH_3O_2] \gtrsim 10^{11}$ molecule $cm^{-3}$ with the present instrument resolution time (0.15 s) at 80 mbar $He:O_2 = 3:1$ and 100 mbar of air.

      The relatively long ring-down times achieved here require the lasers to be blocked for several ms during which the full exponential ring-down is measured. This imposes an upper limit to the ring-down rate. The achieved rate is significantly smaller (6.5 Hz on average) for the following reasons. The width of the resonances of the optical cavity is of the order of 1

kHz, much narrower than the laser linewidth. This makes the injection of light into the cavity inefficient. Reducing the laser linewidth, e.g. with optical feedback techniques, could significantly increase the injection efficiency and the ring-down rate. Moreover, the resonance frequencies jitter and drift due to the unavoidable vibrations associated with the operation of the HIRAC chamber. The cavity length was actively modulated in order to repeatedly force coincidence of laser and resonance frequency. Due to the poor injection efficiency mentioned above, however, not every coincidence resulted in a ring-down

event. Furthermore, a significant fraction of the ring-down events has to be discarded because of the passage of dust particles, moved around by the fans within the chamber, through the cavity axis. The use of an additional optical filter to cut-off the 254 nm light from the background of the CRDS measurements is expected to improve the CRDS sensitivity if the 254 nm lamps are used in HIRAC. The CRDS sensitivity could be further improved by mounting the cavity mirrors along the HIRAC length, which would result in a cavity of about 2 m length containing $CH_3O_2$ radicals, and, hence above the current 1.4 m length.

Although the origin band centred at 7388 $cm^{-1}$ is about three times stronger than the methyl torsional band at 7488 $cm^{-1}$ (Pushkarsky et al., 2000;Chung et al., 2007), the latter was targeted because absorption due to water vapour is between one and three orders of magnitude weaker there (assuming 1% v/v, atmospheric pressure) (Gordon et al., 2017).

### 3.4 Intercomparison of CRDS and FAGE $CH_3O_2$ measurements

All the intercomparison measurements have been performed at 7488 $cm^{-1}$, where the $CH_3O_2$ cross section was determined using CRDS (Sect. 3.2). For the measurements at 80 mbar $He:O_2$ (3:1) and 100 mbar $N_2:O_2$ (4:1), $CH_3O_2$ was generated either from the photolysis of acetone at 254 nm in the presence of $O_2$ or from the photolysis of $Cl_2$ using UV black lamps in the presence of $CH_4/O_2$. At 1000 mbar of synthetic air, the overlap of the methane absorption lines due to the pressure broadening resulted in a significant $CH_4$ absorption over the range from 7486–7491 $cm^{-1}$ in the background of the $CH_3O_2$ measurements.

Therefore, all the measurements at 1000 mbar have been carried out using the photolysis of acetone/$O_2$ at 254 nm. The data recorded by CRDS using acetone/$O_2$ were more scattered than the CRDS data recorded using $Cl_2/CH_4/O_2$ for the reasons discussed above (see Figs. 6a and 8a in comparison with Fig. 7a) and were the main contributors to the scatter on $[CH_3O_2]_{CRDS}$ in the correlation plots (Figs. 6b, 7b and 8b below). There was less signal noise present in the FAGE measurements, where the most significant source of noise is the shot noise (Poisson noise), which increases with the number of photons counted by the

detector (Figs. 1, S3 and S4) and results in a scatter on the FAGE data growing with $[CH_3O_2]$ in Figs 6a, 7a and 8a.

      As the acquisition rate of CRDS (6.5 Hz in average) differed compared to the FAGE acquisition rate (in the range 1–10 Hz) the comparison data were averaged to enable comparison of $[CH_3O_2]$ by the two instruments at the same moments of time.

The averaging interval of time was chosen in the range 3–5 s depending on the comparison measurement to average at least 10 ring-down events over each time interval as the CRDS data were filtered to exclude outliers caused by dust particles passing through the light beam trapped in the optical cavity and the number of encountered 'dust events' varied from one experiment to another.

$CH_3O_2$ was generated over a range of concentrations, $2–26 \times 10^{10}$ molecule $cm^{-3}$ at 80 mbar of He + $O_2$ mixture, $2–60 \times 10^{10}$ molecule $cm^{-3}$ at 100 mbar of synthetic air and $2–30 \times 10^{10}$ molecule $cm^{-3}$ at 1000 mbar of synthetic air. The comparison involved both periods with lamps on where the concentration of $CH_3O_2$ was changing slowly, and where the lamps were turned off and the rapid decay of $CH_3O_2$ was observed. Figures 6a, 7a and 8a show examples of time-resolved $CH_3O_2$ concentrations where the lamps were turned on and off. CRDS absorption coefficients were converted into concentrations using the absorption

cross section determined by studying the second-order recombination kinetics, $\sigma(7488\ cm^{-1}) = (1.49 \pm 0.19) \times 10^{-20}\ cm^2$ $molecule^{-1}$ (Sect. 3.2). The FAGE signals were converted into $[CH_3O_2]$ using the sensitivity factor derived from the analysis of the temporal decays of $CH_3O_2$ at 80 mbar of He + $O_2$ mixture and 100 mbar of air, $(3.83 \pm 0.50) \times 10^{-9}$ counts $cm^3$ $molecule^{-1}$ $s^{-1}\ mW^{-1}$ and $(2.80 \pm 0.37) \times 10^{-9}$ counts $cm^3$ $molecule^{-1}\ s^{-1}\ mW^{-1}$, respectively. The data in the correlation plots of the $CH_3O_2$ concentrations determined by FAGE ($y$-axis) and CRDS ($x$-axis) (Figs. 6b, 7b and 8b) were fitted using an orthogonal distance

linear regression fit (Boggs et al., 1987), which accounts for errors in both the $y$- and $x$-directions. The gradient of the correlation plot of the $CH_3O_2$ concentrations determined by FAGE ($y$-axis) and CRDS ($x$-axis) at 80 mbar of He + $O_2$ (Fig. 6b) is $1.03 \pm 0.05$, showing an overall level of agreement within 3%. The gradient of the correlation plot of the $CH_3O_2$ concentrations determined by FAGE ($y$-axis) and CRDS ($x$-axis) at 100 mbar of air (Fig. 7b) is $0.95 \pm 0.02$, showing an overall level of agreement within 5%.

At 1000 mbar of air, the FAGE signal observed in HIRAC could be calibrated in one of two ways, either via the photolysis of water vapour to generate OH followed by reaction with $CH_4$ to form $CH_3O_2$, or via the kinetic analysis of second order temporal decays of $CH_3O_2$. The conversion of the FAGE signals into $[CH_3O_2]$ at 1000 mbar air for the intercomparison with CRDS shown in Figs. 8a and 8b was based on the average of the results of the water vapour calibration method and the kinetic decay calibration method, which gives $\overline{C}_{CH_3O_2} = (9.81 \pm 2.03) \times 10^{-10}$ counts $cm^3$ $molecule^{-1}\ s^{-1}\ mW^{-1}$ (Sect. 2.2.2)). The

gradient of the overall correlation plot (Fig. 8b) using $\overline{C}_{CH_3O_2}$ is $1.09 \pm 0.06$, showing agreement to within 9%. Figure S7 in the Supplementary Information shows separately the two correlation plots obtained using the sensitivities from the two methods of calibration for FAGE: $C_{CH_3O_2} = (8.03 \pm 1.37) \times 10^{-10}$ counts $cm^3$ $molecule^{-1}\ s^{-1}\ mW^{-1}$ (water vapour calibration method) and $C_{CH_3O_2} = (1.16 \pm 0.15) \times 10^{-9}$ counts $cm^3$ $molecule^{-1}\ s^{-1}\ mW^{-1}$ (second order kinetic decay method). The gradients of the two linear fits are: $1.35 \pm 0.07$ (water vapour calibration) and $0.92 \pm 0.05$ (kinetic method of calibration). Therefore, a

significantly better agreement (within 8%) was obtained by using the kinetic method for the calibration of FAGE compared with using the water vapour method for calibration of FAGE (35% agreement). Better agreement is expected when using the kinetic method to calibrate FAGE, as this is the same method used to determine the absorption cross section and hence calibrate the CRDS method, and the intercomparison is not affected by any error in the rate coefficient, $k_{obs}$, for the $CH_3O_2$ self-reaction. We consider that the main contribution to the discrepancy in $C_{CH3O2}$ values obtained by the two methods of calibration derives

from an overestimation of the reported value of the observed rate coefficient for the $CH_3O_2$ self-reaction, $k_{obs} = (4.8 \pm 0.6) \times 10^{-13}$ molecule$^{-1}$ $cm^3$ $s^{-1}$ ($1\sigma$ error) at 298 K (Atkinson et al., 2006). In a subsequent paper we will report a revised $k_{obs}$, which will bring the two methods of calibration into agreement.


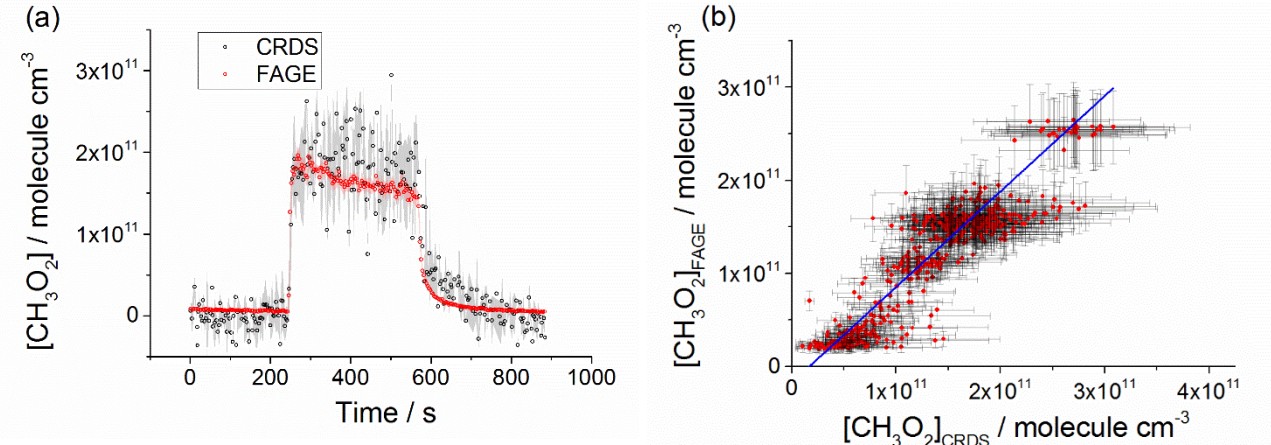

**Figure 6**. (a) Comparison of CH$_3$O$_2$ measurement at 80 mbar He:O$_2$ (3:1) where the lamps were turned on at $t \sim 250$ s for $\sim 5$ min to generate CH$_3$O$_2$ and then turned off again. The measurement by FAGE is shown in red and the measurement by CRDS is plotted in black. CH$_3$O$_2$ radicals were generated using the 254 nm photolysis of (CH$_3$)$_2$CO ($8.8 \times 10^{14}$ molecule cm$^{-3}$). The $1\sigma$ statistical errors generated by the data averaging are shown as grey (CRDS) and red (FAGE) shadows. (b) Correlation plot at 80 mbar He:O$_2$ (3:1) combining the data obtained using acetone/O$_2$/254 nm lamps with the data generated using Cl$_2$/CH$_4$/O$_2$/UV black lamps. [CH$_3$O$_2$] measured by FAGE is plotted against [CH$_3$O$_2$] measured by CRDS. The linear fit to the data generates a gradient of $1.03 \pm 0.05$ and an intercept of $(-1.7 \pm 0.5) \times 10^{10}$ molecule cm$^{-3}$. The linear fits were generated using the orthogonal distance regression algorithm; fit errors at $2\sigma$ level. In both panels [CH$_3$O$_2$]$_{FAGE}$ was determined using a calibration factor of $3.83 \times 10^{-9}$ counts cm$^3$ molecule$^{-1}$ s$^{-1}$ mW$^{-1}$ and [CH$_3$O$_2$]$_{CRDS}$ was calculated using a cross section of $1.49 \times 10^{-20}$ cm$^2$ molecule$^{-1}$. Each point is a value averaged over 3 s.

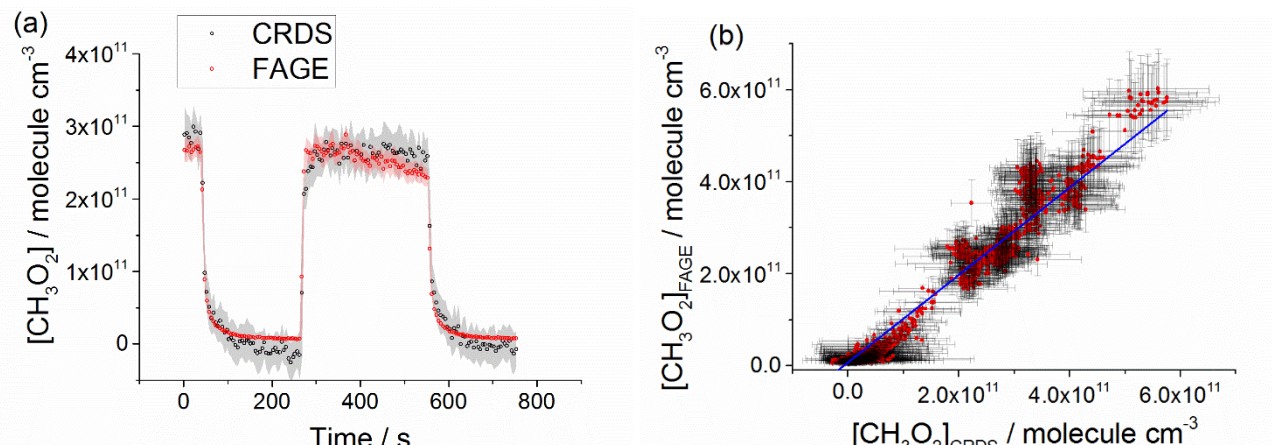

**Figure 7**. Comparison of CH$_3$O$_2$ measurement (a) and the correlation plot at 100 mbar N$_2$:O$_2$ (4:1) (b). In both figures [CH$_3$O$_2$]$_{FAGE}$ was computed using a calibration factor of $2.80 \times 10^{-9}$ counts cm$^3$ molecule$^{-1}$ s$^{-1}$ mW$^{-1}$ and [CH$_3$O$_2$]$_{CRDS}$ was determined using a cross section of $1.49 \times 10^{-20}$ cm$^2$ molecule$^{-1}$. Each point is a value averaged over 5 s. Figure (a) shows the measurement by FAGE (red) and the measurement by CRDS (black) where the CH$_3$O$_2$ radicals were generated by the photolysis of Cl$_2$ ($2.5 \times 10^{15}$ molecule cm$^{-3}$) in the presence of CH$_4$ ($2.4 \times 10^{16}$ molecule cm$^{-3}$) and O$_2$. The UV black lamps were alternately switched on and off: the lamps were turned off at $t \sim 40$ s and then turned on at $t \sim 250$ s for $\sim 5$ min before being switched off again. The $1\sigma$ statistical errors generated by the data averaging are shown as grey (CRDS) and red (FAGE) shadows. Figure (b) combines the data obtained using acetone/O$_2$/254 nm lamps with the data generated using Cl$_2$/CH$_4$/O$_2$/UV black lamps. [CH$_3$O$_2$] measured by FAGE is plotted versus [CH$_3$O$_2$] measured by CRDS. The linear fit to the data is obtained using the orthogonal distance regression algorithm and results in a gradient of $0.95 \pm 0.02$ and an intercept of $(7.0 \pm 0.4) \times 10^9$ molecule cm$^{-3}$; fit errors given at $2\sigma$ level.

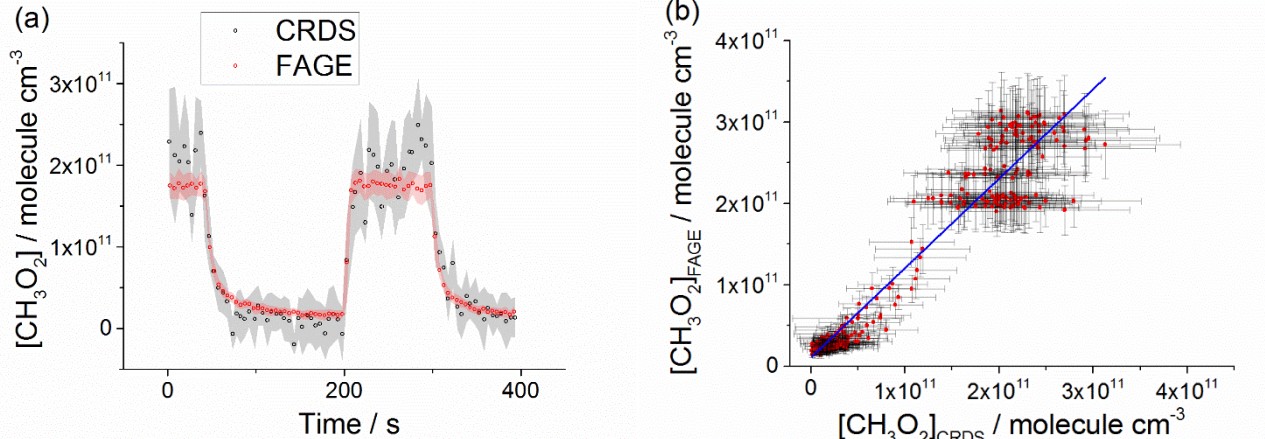

**Figure 8.** (a) Comparison of $CH_3O_2$ measurement at 1000 mbar of synthetic air where the lamps were turned off at $t \sim 40$ s and then on at $t \sim 200$ s for $\sim 2$ min before being switched off again. The measurement by FAGE is shown in red and the measurement by CRDS is plotted in black. $CH_3O_2$ radicals were generated using the 254 nm photolysis of $(CH_3)_2CO$ ($8.8 \times 10^{14}$ molecule $cm^{-3}$). The $1\sigma$ statistical errors generated by the data averaging are shown as grey (CRDS) and red (FAGE) shadows. (b) Correlation plot of all the data generated at 1000 mbar of air. $[CH_3O_2]$ measured by FAGE is plotted against $[CH_3O_2]$ measured by CRDS. The linear fit to the data is generated using the orthogonal distance regression algorithm and results in a gradient of $1.09 \pm 0.06$ and an intercept of $(1.1 \pm 0.3) \times 10^{10}$ molecule $cm^{-3}$; fit errors given at $2\sigma$ level. In both panels $[CH_3O_2]_{FAGE}$ was determined using a calibration factor of $9.81 \times 10^{-10}$ counts $cm^3$ molecule$^{-1}$ s$^{-1}$ mW$^{-1}$ and $[CH_3O_2]_{CRDS}$ was calculated using a cross section of $1.49 \times 10^{-20}$ cm$^2$ molecule$^{-1}$. Each point is a value averaged over 5 s.

## 4 Conclusions

An intercomparison between the recently developed indirect method for the measurement of the $CH_3O_2$ radicals using Fluorescence Assay by Gas Expansion (FAGE) (Onel et al., 2017b) and the direct Cavity Ring-Down Spectroscopy (CRDS) method has been performed within the Leeds Highly Instrumented Reactor for Atmospheric Chemistry (HIRAC). CRDS detected $CH_3O_2$ by using the $A \leftarrow X$ ($\nu_{12}$) electronic transition at 7488 cm$^{-1}$. The $CH_3O_2$ radical was generated from the photolysis of mixtures of either $Cl_2/CH_4/O_2$ or acetone//$O_2$ at room temperature and three total pressures, 80 mbar of He:$O_2$ = 3:1 and 100 and 1000 mbar of $N_2$:$O_2$ = 4:1, and was measured simultaneously using the two methods.

At all pressures FAGE was calibrated using the kinetics of the $CH_3O_2$ second order decay by self–reaction. At 1000 mbar the conventional 185 nm photolysis of water vapour in the presence of excess of $CH_4$ and $O_2$ was used to calibrate FAGE in addition to the kinetic method. The two calibration methods have overlapping error limits at $2\sigma$ level (34% for the water vapour photolysis method and 26% for the kinetic method) as it has been found previously (Onel et al., 2017b). The difference between $C_{CH_3O_2}$(water vapour method) and $C_{CH_3O_2}$(kinetic method) has been discussed in detail previously (Onel et al., 2017b). In the case of $HO_2$, a very good agreement (difference within 8%) between $C_{HO_2}$(water vapour method) and $C_{HO_2}$(kinetic method) was obtained previously (Onel et al., 2017a;Winiberg et al., 2015), which suggests that the production of OH and $HO_2$ from the photolysis of water vapour in air can be quantified robustly. We consider it unlikely that there is a significant error in the fraction of OH which is converted to $CH_3O_2$ upon the addition of methane. We consider instead that the discrepancy between the two calibration methods is due to an overestimation of the reported value of $k_{obs}$ for the $CH_3O_2$ self-reaction (Atkinson et al., 2006); the two methods of calibrations agree if $k_{obs}$ is reduced by 25–30%, which is close to the reported $2\sigma$ uncertainty in the rate coefficient (Atkinson et al., 2006). The average value of the sensitivity factor obtained from the two calibration methods, $\bar{C}_{CH_3O_2} = (9.81 \pm 2.03) \times 10^{-10}$ counts $cm^3$ molecule$^{-1}$ s$^{-1}$ mW$^{-1}$, corresponds to a limit of detection (*LOD*) for $CH_3O_2$ of $1.18 \times 10^8$ molecule $cm^{-3}$ for a *S/N* of 2 and 60 s averaging period. The FAGE sensitivity factor increased by $\sim 3$ times by decreasing the pressure in the FAGE detection cell (from 3.3 mbar, corresponding to a total HIRAC pressure of 1000, to 0.9 mbar, corresponding to a total chamber pressure of 100 or 80 mbar).

The $CH_3O_2$ absorption cross section at 7488 cm$^{-1}$ at 100 mbar of air and 80 mbar of He:O$_2$ = 3:1 was determined using the kinetics of the $CH_3O_2$ second order decays: $\sigma_{CH3O2} = (1.49 \pm 0.19) \times 10^{-20}$ cm$^2$ molecule$^{-1}$. No change in the shape of the $CH_3O_2$ spectrum with pressure was found from the reduced pressures (100 mbar of air and 80 mbar of He:O$_2$ = 3:1) to 1000 mbar of air, showing that $\sigma_{CH3O2}$ is almost independent of pressure. For a time averaging of 60 s the calculated CRDS *LOD* using the Allan-Werle deviation plots and $\sigma_{CH3O2}$ is around $8 \times 10^9$ molecule cm$^{-3}$ using acetone/O$_2$/254 nm at all operating pressures and $6 \times 10^9$ molecule cm$^{-3}$ using CH$_4$/Cl$_2$/black lamps at the reduced pressures.

The FAGE–CRDS intercomparison used measurements of $CH_3O_2$ under steady-state conditions (photolysis lamps on) as well as rapid decays in [$CH_3O_2$] (lamps switched off) to cover large concentration ranges: 2–26 × 10$^{10}$ molecule cm$^{-3}$ at 80 mbar of He + O$_2$ mixture, 2–60 × 10$^{10}$ molecule cm$^{-3}$ at 100 mbar of air and 2–30 × 10$^{10}$ molecule cm$^{-3}$ at 1000 mbar of air. A good agreement between [$CH_3O_2$]$_{FAGE}$ and [$CH_3O_2$]$_{CRDS}$ was obtained under all conditions as shown by the gradient of the correlation plots: 1.03 ± 0.05 at 80 mbar He/O$_2$, 0.95 ± 0.02 at 100 mbar air and 1.09 ± 0.06 at 1000 mbar air (using an average of the sensitivity factors for the two FAGE calibration methods). The study provides a validation for the indirect FAGE method for $CH_3O_2$ measurements, in agreement with the previous FAGE validation for HO$_2$ measurements (Onel et al., 2017a).

*Data availability*. Data presented in this study are available from the authors upon request (chmlo@leeds.ac.uk and d.e.heard@leeds.ac.uk).

*Competing interests*. The authors declare that they have no conflict of interest.

**Acknowledgements**

This work has received funding from the Natural Environment Research Council (NERC grant number NE/M011208/1), the National Centre for Atmospheric Science and the European Union's Horizon 2020 research and innovation programme through the EUROCHAMP-2020 Infrastructure Activity under grant agreement No 730997. AB thanks to NERC for a studentship awarded in the framework of the SPHERES doctoral training programme (NE/L002574/1). The authors thank Christa Fittschen for helpful discussions on the absorption cross section of $CH_3O_2$.

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
