# Peer review of "An intercomparison of CH3O2 measurements by Fluorescence Assay by Gas Expansion and Cavity Ring–Down Spectroscopy within HIRAC (Highly Instrumented Reactor for Atmospheric Chemistry)"

_Atmospheric Measurement Techniques, 2019_

## Author Comment (AC1) · 20 Nov 2019

The last sentence of the separate abstract posted in the Discussion Forum was not correct. This sentence is corrected below. The abstract is correct in the Discussion Paper (pdf).

At 1000 mbar of synthetic air the correlation plot of [CH3O2]FAGE against [CH3O2]CRDS gave a gradient of 1.09 $\pm$ 0.06. At 100 mbar of synthetic air the gradi-

ent of the FAGE – CRDS correlation plot had a value of 0.95 $\pm$ 0.02 and at 80 mbar of 3:1 He:O2 mixture the correlation plot gradient was 1.03 $\pm$ 0.05. These results provide a validation of the FAGE method to determine concentrations of CH3O2.

---

## Referee Comment (RC1) · Anonymous Referee #1 · 15 Dec 2019

Review of "An intercomparison of $CH_3O_2$ measurements by Fluorescence Assay by Gas Expansion and Cavity Ring-Down Spectroscopy within HIRAC (Highly Instrumented Reactor for Atmospheric Chemistry) by Onel et al.

This paper reports on laboratory measurements of methylperoxy radicals ($CH_3O_2$) using two different instruments (FAGE and CRDS) and with the radicals generated by two different sources. Instruments were calibrated primarily by the observations of the second order decay of $CH_3O_2$, so this procedure is discussed in depth. The paper is well organized and reports the important features of the instruments, the experimental setup and the comparison of the measurements.

While the development of techniques for measurement of methylperoxy radicals is important, as is comparison of the different methods. It is unfortunate that the detection limits for the CRDS method are so large, and obviously, it will not be possible to apply this method to atmospheric measurements. Still laboratory studies of radical kinetics and mechanisms is possible. Also, other cavity approaches may improve the performance of this approach (CAPS, IBBEAS).

This paper is suitable for publication, but this reviewer would like the authors to consider some general and specific comments in the preparation of their final manuscript.

General comments.

Regarding the calibration of the FAGE, it was surprising that the water vapour photolysis method was not used at all conditions studied. This instrument has been deployed on aircraft, so sensitivities for OH and $HO_2$ were surely determined as a function of sample pressure. It is straightforward to make use of this calibration procedure at reduced pressures for methylperoxy radicals. The paper would be greatly improved by performing such calibrations for the two other chamber conditions. On the other hand, since there is a systematic difference between the two methods of calibrations at 1000 torr, and that the authors favor the water vapour photolysis method (indicating that perhaps the rate coefficient for methylperoxy radical decay, $k_{obs}$, should be reduced by 25-30%), it appears to this reviewer that perhaps the FAGE calibration factors for the 80 mbar and 100 mbar should be scaled by the ratio of the calibration factors for the two methods performed at 1000 mbar. If the rate coefficient is indeed in error, the CRDS calibrations should also be adjusted by the same factor. As stated in the paper, though, if both instruments rely on methylperoxy radical decay for their calibration, then the accuracy of the value for $k_{obs}$ does not matter, if the same value is used for both. Perhaps some more discussion of this in the introduction of the decay method should be included in the paper. It was also not obvious from the paper what $k_{obs}$ referred to, so suggest adding some text to indicate that it refers to the effective rate coefficient for second order methylperoxy radical decay that includes contributions from methylperoxy radicals reacting with $HO_2$, which is produced from R5b followed by the rapid reaction of methoxy radicals with oxygen. It is possible that this is what lines 31-34 on page 6 were trying to say, but it was not clear to this reviewer. Suggest rewriting this text and/or adding more information. Note that without radical wall loss, $k_{obs} = k_{5a}+2k_{5b}$. It would be useful to add discussion on the impact of $HO_2$ wall loss on $k_{obs}$ both in the present experiments, and in those used by IUPAC to arrive at their kinetic recommendations.

Regarding the three conditions for the intercomparison: why were these selected? Would it have made sense to perform experiments in air at a variety of pressures (say five) between 100 and 1000 mbar? It is not apparent why helium/oxygen mixtures were used. Suggest adding some more discussion of the reasoning for the selection of chamber conditions for the intercomparisons.

Clearly, analytical techniques for methylperoxy radicals are going to be useful in application to reactions of these radicals, such as R5 and R12. This paper states that studies of R5 kinetics will be

reported in another paper. The authors may wish to consider publishing the other paper first, since such results could have direct bearing on the results of this paper.

Specific comments.

Page 4, line 24, 31-33. Very high concentrations of chlorine, methane and acetone were used in these experiments. Have you verified that these high amounts do not affect the performance of the instruments through interferences or artifacts? If so, suggest adding a discussion of the tests that were performed.

Page 4, line 38 to page 5, line5. Suggest giving the transit times in addition to the distances in describing the sample moving through the FAGE.

Page 4, line 11-12. Suggest giving units for $C_{CH3O2}$ and $S_{CH3O2}$ factors.

Page 6, equations 3, 4, and 5. Suggest using $\Delta t$ instead of $t$ in these equations.

Page 6, line 26. Suggest rewording "…does not stop the decay analysis…".

Page 6, lines 30-34. Suggest rewording (mentioned earlier) this discussion.

Page 7, lines 1 and 3. Suggest adding discussion of the fitting procedure used in this paper, both for the radical signal decays and the instrument comparisons.

Page 7, line 12. It is stated that "…wall losses are very small and can be neglected." Suggest adding an upper limit for the wall loss of $CH_3O_2$ and for $HO_2$ (since this bears on the decay of $CH_3O_2$).

Page 7, line 21-22. It appears that the exponent for the sensitivity should be "-9" rather than "-10" as shown.

Page 8, line 9. Suggest "…S2 and S3, respectively."

Page 8, line 30. Does the large amount of ozone affect the instrument performance?

Page 10, line 14. Is it not possible to keep the radical concentration stable for more than 5 minutes? Perhaps this sentence needs rewording.

Page 10, line 35. Suggest rewording "…potential small difference in wavelength compared to λ in the spectrum…". The meaning is not clear as written.

Page 12, line 28. It is not clear where the term "Allan-Werle deviation plots" originated. As this reviewer understands it, the original term was Allan variance, which was extended to include Peter Werle's name after his contribution of suggesting its use in analysis of tunable diode laser spectroscopy performance. After his death, it was suggested that the term be changed to Allan-Werle variance. Suggest this term be used here.

Page 13, line 7. Suggest adding some explanation why the acquisition rate was only 6.5 Hz. Is it not possible to have 1000 or more ring-down events per second? Perhaps give typical ring-down times. This is mentioned on page 14, line 20. Perhaps indicate how much the frequency could be increased.

Page 13, line 15. Could additional optical filters be added to minimize the impact of the 254 nm photolysis radiation?

Page 15, lines 9 and 11. While "gradient" to describe the slope of a linear fit is technically correct, usually the term "slope" is used.

Page 15, line 25. Suggest "…and hence calibration of the CRDS…".

Page 15, line 26. Suggest "…and the intercomparison is not affect by error in the rate coefficient…".

Page 16, Figure 6(a). The decay of the CRDS and FAGE signals do not show the same temporal behavior. It appears there is a low pass filtering of the CRDS signals. Suggest discussing this. "Linear fit" and "orthogonal distance algorithm" are mentioned in the caption. This should be discussed in the text, with appropriate references. Also include information how the fit errors were determined. In the caption suggest "Each point is a value averaged over 3 seconds." Same suggestion for captions of Figure 7 and 8.

Page 16-17, Figures 6, 7, and 8. The 80 mbar data are averaged for 3 seconds, while the 100 mbar and 1000 mbar data are averaged for 5 seconds. Suggest discussing the logic for selecting various averaging times in the text. For the intercomparison, would it not be better to average for 1 minute or more, and generate different radical concentrations by adjusting the concentrations of precursors and/or the lamp intensity, rather than use the decays to achieve different concentrations? Suggest discussing the logic of the experimental design in the body of the document.

Page 17, Figure 8 caption. Suggest "The measurements by FAGE are shown in red and the measurements by CRDS are plotted in black."

Page 17, line 33. Suggest "…FAGE detection cell (from 3.3 to 0.9 mbar when sampling from a pressure of 1000 mbar." It is not clear why three pressures are listed, or what the "respectively" refers back to. Suggest reworking this last sentence.

Page 18-19. For future reference, it is much easier for readers if the references are formatted as "hanging" paragraphs.

---

## Referee Comment (RC2) · Anonymous Referee #2 · 28 Dec 2019

I find this manuscript well written and with valuable content. Its purpose appears principally to validate the indirect FAGE technique for quantitative measurement of $CH_3O_2$ by calibrating it with direct measurements using CRDS. For the most part, this goal is successfully accomplished, at least for the given FAGE apparatus.

I am generally in agreement with referee 1's comments so I will not repeat them. However, I would note that there are a few instances for which a persuasive argument can be made for maintaining the original text.

There is one area where I would ask for a clearer, more detailed presentation. I'll place my comments in terms of an equation that I consider necessary, but which has been omitted from the text. The equation relates the absorption to the desired results,

$$\alpha_t(\nu) = N \, \sigma(\nu) \, L$$

$\alpha_t(\nu)$ Is defined by equation 8 (or more precisely by equation 9) and is determined directly from the experimental observations. The N is the concentration in the vibrationless level of the ground state of $CH_3O_2$ which is the object of the experiment. The $\sigma(\nu)$ is the cross-section for absorption at a frequency $\nu$.

The $\sigma (\nu)$ Is obviously critical to the determined value of N, yet I find the basis for the value used rather vague. Why did the authors measure absorption at 7488 cm$^{-1}$ when Fittschen(2019) report $\sigma (\nu)$ at 7489 cm$^{-1}$? Given the results in figure 3, why do the authors consider the cross sections at the two wavelengths to be equal? Finally it would be helpful to have stated that the cross-section used is not the more standard integrated cross-section used by HITRAN and other databases.

The quantity, L, the length also requires further explanation. It should not refer to the 1.4 M value of the mirror separation in figure 2. (Morover, the comments on p14, line20-21 that one can improve the sensitivity by increasing the mirror separation, are incorrect.) The appropriate value of L is the effective path length of the sample, taking into account the diminishing concentration of $CH_3O_2$ near to and into the extensions that support the mirrors.

With the indicated modifications, I believe the manuscript should be published. I do however note referee number 1's comments about the desirability of publishing first the paper describing a new value for the rate constant for the self-destruction of $CH_3O_2$. Nonetheless I recognize that this may prove practically

difficult and would not place that as a stipulation for publishing the present paper.

---

## Referee Comment (RC3) · Anonymous Referee #3 · 13 Jan 2020

This manuscript reports the calibration of a laser-induced fluorescence instrument (FAGE design) dedicated to the measurement of methyl peroxy radicals using two calibration approaches: 1/ water-photolysis at 184.9 nm and ambient pressure in presence of CH4 and 2/ kinetic decays of CH3O2 from its self-reaction. The latter, performed in the Leeds HIRAC chamber, allows calibrating the LIF instrument at ambient and lower pressures (80 and 100 mbar in this study). The results showed that at atmospheric pressure the two calibration approaches agree within 30%. While this difference is

within the combined uncertainty of the 2 approaches, the authors believe that it is due to an overestimation of the CH3O2+CH3O2 rate constant recommended by IUPAC.

The LIF instrument was then compared to a CRDS by generating CH3O2 in HIRAC at low (80 and 100 mbar) and ambient (1000 mbar) pressure. The authors report a good agreement between the 2 instruments with correlation slopes close to unity (within 10%) and indicate that this study validates the use of FAGE for CH3O2 measurements.

There is currently a need for the development and the validation of new techniques capable of measuring speciated organic peroxy radicals and this study provides additional confidence in the use of the FAGE technique for the measurement of one of the most abundant organic peroxy radicals in the atmosphere. This work will be of interest for the scientific community. The manuscript is well written, clear and concise, and this reviewer recommends publication in AMT once the following minor comments have been addressed.

Minor comments:

P2-P3: Several techniques are discussed for the measurement of peroxy radicals. The authors should also briefly discuss the use of chemical ionization mass spectrometry methods such as published in Noziere and Hanson (2017), Noziere and Vereecken (2019), Hansel et al. (2018), Jokinen et al. (2014), etc.

P4 L16 & P5 L19: CH4 is used during calibration experiments as a precursor for CH3O2 and is added in the Water-photolysis calibrator and the HIRAC chamber at concentrations as high as 2.5E17 molecule/cm3. Can the authors comment on the potential impact of CH4 on the quenching of CH3O in the detection cell?

P7 Eq. 6: Please define kloss

P6 L9-13 & P12 L4-6: What were the fitted values for kloss? Are the values inferred from the two experiments consistent with each other?

P6 L22: "1E-10" should read "1E-9"

P15 L9-12 & L17-18 & L21-22: The authors show that correlation plots between FAGE and CRDS exhibit slopes that are close to unity. However, the y-intercepts of the regression lines are not discussed. Were the intercepts not statistically significant?

P16 Figures 6-7: When the lamps are turned off, (1) the CRDS measurements seem to decrease to lower values than FAGE and (2) the FAGE measurements seem to reach a plateau more rapidly than the CRDS. Could the authors comment on this?

References:

Nozière, B. and D. R Hanson (2017), Speciated Monitoring of Gas-Phase Organic Peroxy Radicals by Chemical Ionization Mass Spectrometry: Cross-Reactions between CH3O2, CH3(CO)O2, (CH3)3CO2, and c‑C6H11O2, J. Phys. Chem. A, 121, 8453-8464.

Noziere, B., & Vereecken, L. (2019). Direct Observation of Aliphatic Peroxy Radical Autoxidation and Water Effects: an Experimental and Theoretical Study. Angew. Chem. Int. Ed., doi: 10.1002/ange.201907981.

Hansel, A., Scholz, W., Mentler, B., Fischer, L., & Berndt, T. (2018). Detection of RO2 radicals and other products from cyclohexene ozonolysis with NH4+ and acetate chemical ionization mass spectrometry. Atmos. Environ., 186, 248-255.

Jokinen T, Sipilä M, Richters S, et al. Rapid autoxidation forms highly oxidized RO2 radicals in the atmosphere. Angewandte Chemie (International ed. in English). 2014 Dec;53(52):14596-14600. DOI: 10.1002/anie.201408566.

---

## Author Comment (AC2) · 20 Feb 2020

**Author response to anonymous referee #1 on "An intercomparison of CH₃O₂ measurements by Fluorescence Assay by Gas Expansion and Cavity Ring–Down Spectroscopy within HIRAC (Highly Instrumented Reactor for Atmospheric Chemistry)" by L. Onel et al.**

**General comments**

*Regarding the calibration of the FAGE, it was surprising that the water vapour photolysis method was not used at all conditions studied. This instrument has been deployed on aircraft, so sensitivities for OH and HO2 were surely determined as a function of sample pressure.*

The HIRAC FAGE instrument has not been deployed on an aircraft, although the Leeds aircraft FAGE instrument (Commane et al., 2010) does have fluorescence cells of a similar design to those used in HIRAC for this $CH_3O_2$ work. The sensitivity towards OH and $HO_2$ as a function of pressure of the fluorescence cell was determined for the HIRAC FAGE instrument (Winiberg et al., 2015), and also for the aircraft instrument (Commane et al., 2010). However, in both cases the sample pressure was not changed, rather the pressure in the fluorescence cell was altered by using different sized pinholes to reflect the same pressure as whilst sampling from reduced pressure (either at different altitudes on the aircraft or from HIRAC operating at reduced pressure). For the aircraft instrument there was very little dependence of the sensitivity with cell pressure for the range of altitudes encountered for both OH and $HO_2$ (Commane et al., 2010). For HIRAC, a small increase in the sensitivity for OH was seen, and a larger change for $HO_2$ was seen over the range of cell pressures used (Winiberg et al., 2015), although the experiments were performed at a range of pulse repetition frequencies, and different pumps were used for the aircraft and HIRAC pressure dependencies. This approach, which assumes there is no change in any losses at the pinhole across a changing pressure differential, was validated by another group for an aircraft instrument using different materials for the pinhole (Faloona et al., 2004). In HIRAC, we validated the approach by employing the decay of a hydrocarbon in HIRAC at different pressures in the presence of OH which was measured using FAGE, or that of $HO_2$ by its kinetic decay followed production via the photolysis of HCHO at different pressures, and obtained the same result as using the water vapour calibration method.

References

Commane, R., Floquet, C. F. A., Ingham, T., Stone, D., Evans, M. J., and Heard, D. E.: Observations of OH and $HO_2$ radicals over West Africa, *Atmos. Chem. Phys.*, 10, 8783–8801, 2010.

Winiberg, F. A. F., Smith, S. C., Bejan, I., Brumby, C. A., Ingham, T., Malkin, T. L., Orr, S. C., Heard, D. E., and Seakins, P. W.: Pressuredependent calibration of the OH and HO2 channels of a FAGE HOx instrument using the Highly Instrumented Reactor for Atmospheric Chemistry (HIRAC), *Atmos. Meas. Tech.*, 8, 523-540, 2015.

Faloona, I. C., Tan, D., Lesher, R. L., Hazen, N. L., Frame, C. L., Simpas, J. B., Harder, H., Martinez, M., Di Carlo, P., Ren, X. R., and Brune, W. H.: A laser-induced fluorescence instrument for detecting tropospheric OH and HO2: Characteristics and calibration, *J. Atmos. Chem.*, 47, 139–167, 2004

*It is straightforward to make use of this calibration procedure at reduced pressures for methylperoxy radicals. The paper would be greatly improved by performing such calibrations for the two other chamber conditions.*

In this paper, rather than validating the water vapour method for the calibration of $CH_3O_2$ sensitivity as a function of pressure (using a known OH which is converted to $CH_3O_2$, followed by conversion to $CH_3O$ by NO, and the $CH_3O$ formed detected (Onel et al. 2017b in the list of references of the manuscript), the main aim was to compare two distinct techniques (FAGE and CRDS) for a range of sampling conditions. Also, it is felt that because the sample pressure cannot be reduced in these experiments during the calibration (rather a change in pinhole is used or a change in pumping capacity is used to change the cell pressure, which would change the residence time from sampling to the laser-excitation axis), that additional uncertainties would arise in determining the sensitivity of FAGE to $CH_3O_2$ as a function of pressure using the water vapour calibration method, so it was used at atmospheric pressure to calibrate $CH_3O_2$.

*On the other hand, since there is a systematic difference between the two methods of calibrations at 1000 torr, and that the authors favor the water vapour photolysis method (indicating that perhaps the rate coefficient for methylperoxy radical decay, kobs, should be reduced by 25-30%), it appears to this reviewer that perhaps the FAGE calibration factors for the 80 mbar and 100 mbar should be scaled by the ratio of the calibration factors for the two methods performed at 1000 mbar. If the rate coefficient is indeed in error, the CRDS calibrations should also be adjusted by the same factor. As stated in the paper, though, if both instruments rely on methylperoxy radical decay for their calibration, then the accuracy of the value for kobs does not matter, if the same value is used for both.*

The referee is correct, the same value is used for both and so the accuracy of the value for $k_{obs}$ does not matter. The kinetic method used for the determination of the $CH_3O_2$ absorption cross-section, $\sigma_{CH3O2}$ relies on the value for the rate coefficient of the $CH_3O_2$ self-reaction. However, as the previous studies at a range of pressures typically used the $CH_3O_2$ kinetic decay to determine $\sigma_{CH3O2}$, the kinetic method of calibration was chosen in this work for comparisons of the value of $\sigma_{CH3O2}$ obtained in this work with the values for $\sigma_{CH3O2}$ reported previously.

    The water vapour photolysis method of calibration is a well-established method for FAGE calibration (*vide supra*), routinely used in calibrations at atmospheric pressure. Therefore, both methods of FAGE calibration, the water vapour photolysis method and the kinetic method, were employed in this work at atmospheric pressure. As noted by the referee, a systematic 25–30% discrepancy was found between the FAGE sensitivities factors, $C_{CH3O2}$ obtained by the two methods at atmospheric pressure (this work and Onel et al. 2017b in the list of references of the manuscript). As the water vapour photolysis method is known to be an accurate and reliable method of calibration, the discrepancy at atmospheric pressure would seem to indicate that the value of $k_{obs}$ (Atkinson et al. 2006) is overestimated by 25–30%. This is already discussed in detail in the main text (see lines 24 – 30, page 15 and lines 19 – 30, page 17).

*It was also not obvious from the paper what kobs referred to, so suggest adding some text to indicate that it refers to the effective rate coefficient for second order methylperoxy radical decay that includes contributions from methylperoxy radicals reacting with HO2, which is produced from R5b followed by the rapid reaction of methoxy radicals with oxygen. It is possible that this is what lines 31-34 on page 6 were trying to say, but it was not clear to this reviewer. Suggest rewriting this text and/or adding more information. Note that without radical wall loss, kobs = k5a+2k5b. It would be useful to add discussion on the impact of HO2 wall loss on kobs both in the present experiments, and in those used by IUPAC to arrive at their kinetic recommendations.*

Following the suggestions of the referees the paragraph in the MS corresponding to lines 26-33 on page 6 was changed and now includes both the expression used for the observed rate coefficient in line with IUPAC recommendation: $k_{obs} = k_5(1 + r_{5b})$, where $r_{5b}$ is the branching ratio for the channel R5b, and the rationale behind this expression (see also the response to the other comment regarding line 26 on page 6, see below):

"As each $HO_2$ radical consumes rapidly one $CH_3O_2$ species on the time scale of the reaction R5, the $CH_3O_2$ decay is described by second order kinetics, with $k_{obs} = k_5(1 + r_{5b})$, where $r_{5b}$ is the branching ratio for the channel R5b. By using the IUPAC recommendations (Atkinson et al., 2006): $k_5 = (3.5 \pm 1.0) \times 10^{-13}$ molecule$^{-1}$ cm$^3$ s$^{-1}$ and $r_{5b} = 0.37 \pm 0.06$, a value of $4.8 \times 10^{-13}$ molecule$^{-1}$ cm$^3$ s$^{-1}$ is obtained for $k_{obs}$.

Modelling of the decay process with a variety of $CH_3O_2$ and $HO_2$ concentrations after the lamps were switched off and following the establishment of steady state conditions showed that Eq. (3) was valid within experimental error. With $k_5 = 3.5 \times 10^{-13}$ molecule$^{-1}$ cm$^3$ s$^{-1}$ (Atkinson et al., 2006), a faster observed rate constant (defined by Eq. (3)) was obtained from the model with a value, $4.9 \times 10^{-13}$ molecule$^{-1}$ cm$^3$ s$^{-1}$ consistent with that recommended by IUPAC, $(4.8 \pm 0.6) \times 10^{-13}$ molecule$^{-1}$ cm$^3$ s$^{-1}$ (1$\sigma$ uncertainty; Atkinson et al., 2006). Substituting …"

Details of the experiments which investigated the potential impact of the $HO_2$ wall loss on the value of $k_{obs}$ are included in the response to the comment regarding line 12 on page 7 and described in a paragraph added above Fig. 1, and included in the same response (*vide infra*).

*Regarding the three conditions for the intercomparison: why were these selected? Would it have made sense to perform experiments in air at a variety of pressures (say five) between 100 and 1000 mbar? It is not apparent why helium/oxygen mixtures were used. Suggest adding some more discussion of the reasoning for the selection of chamber conditions for the intercomparisons.*

The pressure of 1000 mbar of synthetic air was chosen to perform measurements under atmospheric conditions. To the best of our knowledge this is the first study of the $CH_3O_2$ absorption feature centred around 7488 cm$^{-1}$ at a relatively high pressure. The previous studies were performed at reduced pressures, in the range ~30 – 200 mbar (see the introduction of the main text). In order to enable comparison with the reported studies of the $CH_3O_2$ spectrum and also test the performance of both instruments (FAGE and CRDS) at reduced pressure part of the experiments were performed at 100 mbar of synthetic air. The pressure of 80 mbar He/$O_2$ mixture was chosen as the most recent reported $CH_3O_2$ absorption spectrum (Faragó et al. 2013) was obtained at reduced pressures (70 and 133 mbar) of He/$O_2$ mixtures. The text

describes all the conditions used previously in the studies of the $CH_3O_2$ absorption spectrum, so already provides some reasoning for why these conditions were chosen.

*Clearly, analytical techniques for methylperoxy radicals are going to be useful in application to reactions of these radicals, such as R5 and R12. This paper states that studies of R5 kinetics will be reported in another paper. The authors may wish to consider publishing the other paper first, since such results could have direct bearing on the results of this paper.*

The purpose of the present study is to provide a validation of the LIF method for $CH_3O_2$ measurements. As mentioned by the referee, even if the value of $\sigma_{CH3O2}$ does rely on the kinetics of the $CH_3O_2$ self-reaction, the $\sigma_{CH3O2}$ value obtained does not affect the results of the FAGE – CRDS intercomparison (as the same value for $k_{obs}$ is used for both) and, hence the validation of the FAGE method. We would like to publish the present results first, which provide a validation of the newly LIF method for $CH_3O_2$ before reporting kinetic studies of $CH_3O_2$ reactions employing the method. A detailed paper describing extensive studies of kinetics of the $CH_3O_2$ self-reaction over a range of temperatures is in preparation. This publication will enable to scale the value of $\sigma_{CH3O2}$ based on any change value of $k_{obs}$ for the $CH_3O_2$ self-reaction, as noted by the referee in the general comments (see above). The current paper is written to be consistent with the detailed kinetics paper to follow. We also mention that referee 2 notes this comment by referee 1, but did not feel that the kinetics paper needed to be published first.

**Specific comments**

*Page 4, line 24, 31-33. Very high concentrations of chlorine, methane and acetone were used in these experiments. Have you verified that these high amounts do not affect the performance of the instruments through interferences or artifacts? If so, suggest adding a discussion of the tests that were performed.*

We respond on this comment first for the CRDS instrument, then for the FAGE instrument.

**CRDS instrument**
The concentrations of the reagents were chosen to generate a range of $CH_3O_2$ concentrations above the detection limit of CRDS at each pressure (Table 2, page 14 in the main manuscript). The molecular chlorine delivery did not result in a change in the ring-down time. However, the methane and acetone delivery led to a decrease in the ring down time due to their absorbance in the range ~7486–7491 $cm^{-1}$ used in the CRDS measurements. The absorption coefficient of acetone in a typical concentration of $\sim 9 \times 10^{14}$ molecule $cm^{-3}$ was measured in the absence of $CH_3O_2$ (before to turn the HIRAC lamps on to generate $CH_3O_2$) to obtain a value of $\sim 8 \times 10^{-9}$ $cm^{-1}$ at 7487.98 $cm^{-1}$. Similar measurements, in the absence $CH_3O_2$ were performed to determine the $CH_4$ absorption coefficient, $\alpha_{CH4}$. For the typical concentrations of $CH_4$, in the range of $(1.2–2.5) \times 10^{16}$ molecule $cm^{-3}$, $\alpha_{CH4, 7487.98\,cm-1} \approx (0.7 - 1.4) \times 10^{-8}$ $cm^{-1}$. The absorption of acetone and methane in the background of the CRDS measurements of $CH_3O_2$ was taken into account in the determination of the $[CH_3O_2]_{CRDS}$. We have modified the wording in the MS as described in the paragraph below, added after line 10, page 10:

"The molecular chlorine delivery did not result in a change in the measured ring-down time. However, he delivery of the methane and acetone reagents led to a decrease in the ring-down time indicating that, in the concentrations delivered to the chamber, methane and acetone absorbed in the wavenumber range used in the present work, ~7486–7491 cm$^{-1}$. An absorption coefficient of ~$8 \times 10^{-9}$ cm$^{-1}$ was measured for [acetone] $\approx 9 \times 10^{14}$ molecule cm$^{-3}$ at the typical measurement point of 7487.98 cm$^{-1}$ (*vide infra*). An absorption coefficient in the range (0.7–1.4) $\times 10^{-8}$ cm$^{-1}$ was determined at 7487.98 cm$^{-1}$ for CH$_4$ in typical concentrations in the FAGE–CRDS intercomparison experiments in the range (1.2–2.5) $\times 10^{16}$ molecule cm$^{-3}$. The background ring-down time $\tau_0$ (Eq. 7) contained the contributions of the reagents, methane or acetone, and was monitored regularly during the experiments by turning off the chamber lamps (*vide supra*)."

**FAGE instrument**
The concentrations of the reagents were a few orders of magnitude smaller in the fluorescence detection cell than in the HIRAC chamber as the gas mixture was sampled into the FAGE instrument through a 1.0 mm diameter pinhole nozzle resulting in the pressure in the FAGE detection cell being a few orders of magnitude lower than the pressure in HIRAC. The concentrations of the reagents were changed with no discernible change in the FAGE sensitivity factor – this is now mentioned in the text, see below.

The lines 39- 40, page 4 in Sect 2.2 were changed to …:

"…The interior of the tube is held at a low pressure (3.3 mbar for a HIRAC pressure, $p_{HIRAC}$ of 1000 mbar of synthetic air and 0.9 mbar for $p_{HIRAC}$ = 100 mbar synthetic air and $p_{HIRAC}$ = 80 mbar mixture of He:O$_2$ = 3:1) and..."

The investigations described below showed that there was no effect of the concentrations of the reagents (Cl$_2$, methane and acetone) on the sensitivity factor of FAGE.

The following text was added after line 17, page 8 in the section 2.2.2:

"As the pressure in the FAGE detection cell was 2-3 orders of magnitude lower than the corresponding pressure in HIRAC (*vide supra* in Sect. 2.2) the concentrations of the reagents (Cl$_2$, methane and acetone) were also 2-3 orders of magnitude lower in the fluorescence cells than the reagent concentrations in HIRAC. However, a potential effect of the reagents (Cl$_2$, methane and acetone) on the FAGE sensitivity factor in the HIRAC experiments was investigated. Two different concentrations of CH$_4$ were used in the kinetic method for FAGE calibration at 80 mbar of He + O$_2$ in HIRAC to find practically the same sensitivity factor: $(3.80 \pm 0.50) \times 10^{-9}$ counts cm$^3$ molecule$^{-1}$ s$^{-1}$ mW$^{-1}$ for $2.5 \times 10^{16}$ molecule cm$^{-3}$ CH$_4$ ($2.8 \times 10^{14}$ molecule cm$^{-3}$ in the fluorescence cell) and $(3.86 \pm 0.50) \times 10^{-9}$ counts cm$^3$ molecule$^{-1}$ s$^{-1}$ mW$^{-1}$ for $2.5 \times 10^{17}$ molecule cm$^{-3}$ CH$_4$ ($2.8 \times 10^{15}$ molecule cm$^{-3}$ in the fluorescence cell).

As shown in Fig. S1 in the Supplement there is a good agreement between the laser excitation scans of CH$_3$O obtained from the CH$_3$O$_2$ generated in HIRAC using the two methods: acetone photolysis and Cl$_2$ photolysis in the presence of CH$_4$ and O$_2$. In addition, a good agreement has been previously found between the laser excitation spectra of CH$_3$O generated using the reaction of CH$_4$ with OH (generated by the 254 nm photolysis of water) in the presence of O$_2$ and directly, through the 254 nm photolysis of CH$_3$OH. Therefore, no effect of the used reagents on the laser excitation spectrum of CH$_3$O was found."

*Page 4, line 38 to page 5, line5. Suggest giving the transit times in addition to the distances in describing the sample moving through the FAGE.*

Line 41, page 4: The value given for the flow rate of the gas sampled through the FAGE pinhole was corrected:

"…on one end of the tube at a rate of ~3 SLM."

A sentence was added in the line 4 at page 5:

" … $CH_3O_2$ measurements detailed here. The $CH_3O_2$ radicals sampled through the FAGE pinhole at 1000 mbar in HIRAC reached the detection region in about 85 ms."

*Page 5, line 11-12. Suggest giving units for $C_{CH3O2}$ and $S_{CH3O2}$ factors.*

The units were included in the text.

*Page 6, equations 3, 4, and 5. Suggest using $\Delta t$ instead of $t$ in these equations.*

$t$ was changed to $\Delta t$ in the equations.

*Page 6, line 26. Suggest rewording "…does not stop the decay analysis…" and page 6, lines 30-34. Suggest rewording (mentioned earlier) this discussion.*

Lines 26-33 on page 6 were reworded:

"As each $HO_2$ radical consumes rapidly one $CH_3O_2$ species on the time scale of the reaction R5, the $CH_3O_2$ decay is described by second order kinetics, with $k_{obs} = k_5(1 + r_{5b})$, where $r_{5b}$ is the branching ratio for the channel R5b. By using the IUPAC recommendations (Atkinson et al., 2006): $k_5 = (3.5 \pm 1.0) \times 10^{-13}$ molecule$^{-1}$ cm$^3$ s$^{-1}$ and $r_{5b} = 0.37 \pm 0.06$, a value of $4.8 \times 10^{-13}$ molecule$^{-1}$ cm$^3$ s$^{-1}$ is obtained for $k_{obs}$.

Modelling of the decay process with a variety of $CH_3O_2$ and $HO_2$ concentrations after the lamps were switched off and following the establishment of steady state conditions showed that Eq. (3) was valid within experimental error. With $k_5 = 3.5 \times 10^{-13}$ molecule$^{-1}$ cm$^3$ s$^{-1}$ (Atkinson et al., 2006), a faster observed rate constant (defined by Eq. (3)) was obtained from the model with a value, $4.9 \times 10^{-13}$ molecule$^{-1}$ cm$^3$ s$^{-1}$ consistent with that recommended by IUPAC, $(4.8 \pm 0.6) \times 10^{-13}$ molecule$^{-1}$ cm$^3$ s$^{-1}$ ($1\sigma$ uncertainty; Atkinson et al., 2006). Substituting …"

*Page 7, lines 1 and 3. Suggest adding discussion of the fitting procedure used in this paper, both for the radical signal decays and the instrument comparisons.*

The fitting algorithm was included in line 2, page 7: "which is then used to fit to the experimental data with $k_{obs}$ fixed to the value recommended by IUPAC for 298 K, $4.8 \times 10^{-13}$ molecule$^{-1}$ cm$^3$ s$^{-1}$, using the Levenberg-Marquardt algorithm."

Lines 20-22, page 7 (caption of figure 1): The value given for $C_{CH3O2}$ was corrected and the used fitting algorithm was included: "The data were fitted to Eq. (5) (excluding the wall loss rate, $k_{loss}$; red line) and Eq. (6) (including $k_{loss}$; blue dashed line) using the Levenberg-Marquardt algorithm. The obtained value for the sensitivity factor was the same by both fits: $C_{CH3O2} = (1.17 \pm 0.04) \times 10^{-9}$ counts cm$^3$ molecule$^{-1}$ s$^{-1}$ mW$^{-1}$."

The discussion of the fitting procedure used in the FAGE-CRDS correlation plots was added to the text (*vide infra*).

*Page 7, line 12. It is stated that "...wall losses are very small and can be neglected." Suggest adding an upper limit for the wall loss of CH3O2 and for HO2 (since this bears on the decay of CH3O2).*

The upper limit for the wall loss rate coefficient of $CH_3O_2$ was added (lines 11-12 on page7): "… the small values extracted for $k_{loss}$ (upper limit of $\sim 1 \times 10^{-5}$ s$^{-1}$) fitting Eq. (6) demonstrates that wall losses can be neglected."

A paragraph regarding investigations into a potential impact of the wall loss of HO$_2$ on the analysis was added above figure 1:

"Modelling the $CH_3O_2$ decays including a wall loss for HO$_2$ in the range of measured values $0.03 - 0.09$ s$^{-1}$ (Onel et al. 2017a in the MS), showed an minor impact of the wall loss of HO$_2$ on $k_{obs}$, i.e. $k_{obs}$ within $98 - 95$ % agreement with the IUPAC preferred value, $(4.8 \pm 0.6) \times 10^{-13}$ molecule$^{-1}$ cm$^3$ s$^{-1}$ ($1\sigma$ uncertainty; Atkinson et al., 2006)."

*Page 7, line 21-22. It appears that the exponent for the sensitivity should be "-9" rather than "-10" as shown.*

The power was changed to "-9".

*Page 8, line 9. Suggest "...S2 and S3, respectively."*

A comma was added after S3.

*Page 8, line 30. Does the large amount of ozone affect the instrument performance?*

No effect of O$_3$ on the instrument was encountered. Note that due to the lower pressure in the FAGE detection cell (3.3 mbar) compared to the pressure in the chamber (1000 mbar) in these experiments $[O_3]_{FAGE} = 8.3 \times 10^{10}$ molecule cm$^{-3}$ for $[O_3]_{HIRAC} = 2.5 \times 10^{13}$ molecule cm$^{-3}$.

*Page 10, line 14. Is it not possible to keep the radical concentration stable for more than 5 minutes? Perhaps this sentence needs rewording.*

The result was revised and the time was changed from 5 min to 10 min.

*Page 10, line 35. Suggest rewording "…potential small difference in wavelength compared to λ in the spectrum…". The meaning is not clear as written.*

Due to the difference in the resolution of the two $CH_3O_2$ spectra – the spectrum obtained in this work and the spectrum reported by Faragó et al. (2013) – it is difficult to realise if there are any slight shifts of the spectrum found by Faragó et al. (2013) relative to the spectrum reported in this study.

Therefore the words "potential small difference in wavelength compared to λ in the spectrum" on line 35, page 10 were removed and therefore, the lines 33 – 35, page 10 are changed to: "The peaks at the top of the spectral feature reported by Faragó et al. (2013) are not reproduced in this work owing to the method of generating the spectrum, which did not allow for a high spectral resolution (Sect. 2.3). Previously Pushkarsky et al. (2000)…"

*Page 12, line 28. It is not clear where the term "Allan-Werle deviation plots" originated. As this reviewer understands it, the original term was Allan variance, which was extended to include Peter Werle's name after his contribution of suggesting its use in analysis of tunable diode laser spectroscopy performance. After his death, it was suggested that the term be changed to Allan-Werle variance. Suggest this term be used here.*

The term "Allan-Werle deviation plots" was changed to "plots of the square root of the Allan-Werle variance", as suggested by the referee. Therefore, the following lines were changed to the text shown below:
- line 28, page 12: "… using plots of the square root of the Allan-Werle variance (Werle et al., 1993; Onel et al., 2017a)…"
- lines 2-3, page 13: "The square root of the Allan-Werle variance, $\sigma_A(n)$, gives an estimate of the error…"
- line 9, page 13 (in the caption of Fig. 5): "…An example of the square root of the Allan-Werle variance of the absorption coefficient at 7488 $cm^{-1}$, $\sigma_A(n)$ as a function of the number of ring-down events averaged, $n$ obtained in the absence of $CH_3O_2$ and in the presence of a typical acetone concentration of $8.8 \times 10^{14}$ molecule $cm^{-3}$ at 1000 mbar." Now the fig. 5 label reads "$\sigma_A(n)/cm^{-1}$" instead of "Allan-Werle deviation $\sigma_A(n)/cm^{-1}$".
- line 18, page 13 was changed to: "…Therefore, separate plots of $\sigma_A(n)$ were constructed…"
- line 2, page 14 (Table 2 title) was changed to: "…from the plots of $\sigma_A(n)$ (Fig. 5 shows an example), …"

*Page 13, line 7. Suggest adding some explanation why the acquisition rate was only 6.5 Hz. Is it not possible to have 1000 or more ring-down events per second? Perhaps give typical ring-down times. This is mentioned on page 14, line 20. Perhaps indicate how much the frequency could be increased.*

The suggested explanation was added to the text. Lines 20-21 on page 14, so the original text: "The CRDS sensitivity could be further improved by increasing the frequency of the ring–down events and using a cavity length above the current 1.4 m length."

were replaced with:

"The relatively long ring-down times achieved here require the lasers to be blocked for several ms during which the full exponential ring-down is measured. This imposes an upper limit to the ring-down rate. The achieved rate is significantly smaller (6.5 Hz on average) for the following reasons. The width of the resonances of the optical cavity is of the order of 1 kHz, much narrower than the laser linewidth. This makes the injection of light into the cavity inefficient. Reducing the laser linewidth, e.g. with optical feedback techniques, could significantly increase the injection efficiency and the ring-down rate. Moreover, the resonance frequencies jitter and drift due to the unavoidable vibrations associated with the operation of the HIRAC chamber. The cavity length was actively modulated in order to repeatedly force coincidence of laser and resonance frequency. Due to the poor injection efficiency mentioned above, however, not every coincidence resulted in a ring-down event. Furthermore, a significant fraction of the ring-down events has to be discarded because of the passage of dust particles, moved around by the fans within the chamber, through the cavity axis.

The CRDS sensitivity could be further improved by mounting the cavity mirrors along the HIRAC length, which would result in a cavity of about 2 m length containing $CH_3O_2$ radicals, and, hence above the current 1.4 m length…"

*Page 13, line 15. Could additional optical filters be added to minimize the impact of the 254 nm photolysis radiation?*

The below sentence was added at the beginning of the paragraph above the section 3.4 (page 14):
"The use of an additional optical filter to cut-off the 254 nm light from the background of the CRDS measurements is expected to improve the CRDS sensitivity if the 254 nm lamps are used in HIRAC. The CRDS sensitivity could be further improved…"

*Page 15, lines 9 and 11. While "gradient" to describe the slope of a linear fit is technically correct, usually the term "slope" is used.*

As linear fits were used in the correlation plots the word "gradient" is adequate and was not changed to "slope".

*Page 15, line 25. Suggest "…and hence calibration of the CRDS…".*

"…and hence calibrate of the CRDS method…" was changed to "…hence calibrate the CRDS method…"

*Page 15, line 26. Suggest "…and the intercomparison is not affect by error in the rate coefficient…".*

"…and the intercomparison is not subject to any error in the rate coefficient…" was changed to "…and the intercomparison is not affected by any error in the rate coefficient…"

*Page 16, Figure 6(a). The decay of the CRDS and FAGE signals do not show the same temporal behavior. It appears there is a low pass filtering of the CRDS signals. Suggest discussing this. "Linear fit" and "orthogonal distance algorithm" are mentioned in the caption. This should be discussed in the text, with appropriate references. Also include information how the fit errors were determined. In the caption suggest "Each point is a value averaged over 3 seconds." Same suggestion for captions of Figure 7 and 8.*

Regarding the comment on Figure 6(a): "*The decay of the CRDS and FAGE signals do not show the same temporal behaviour.*":

   The temporal changes in the concentration of $CH_3O_2$ measured by CRDS and FAGE are in good agreement under all the used conditions, as described in the manuscript. There are only slight discrepancies at longer decay times, which are more evident at reduced pressure (80 mbar of He + $O_2$, Figure 6(a) and 100 mbar of air, Figure 7(a)) than at 1000 mbar of air (Figure 8(a)). In Figures 6(a) and 7(a) the FAGE measurements are levelling down due to the second order kinetics going to a constant value, $[CH_3O_2] = 0$ whereas $[CH_3O_2]_{CRDS}$ continues to go down as some reaction products absorbing at the measuring wavenumber (7488 cm$^{-1}$) are changing with time. This in turn slightly changes the absorption even when the $CH_3O_2$ has reached zero concentration. The better FAGE – CRDS agreement at longer times at 1000 mbar than at 80 and 100 mbar could be due to a greater wall loss of the absorbing products at reduced pressures, where diffusion becomes more significant. However, even at 80 and 100 mbar the discrepancies noticed by the referee are minor and the FAGE – CRDS correlation plots, which incorporate all the temporal decay data show a good agreement under all conditions.

Regarding "*It appears there is a low pass filtering of the CRDS signals*":

The reviewer is right and now the following paragraph was added in line 2, page 10:
"…to extract the ring-down time, $\tau$. Filters were applied to process the ring-down events to exclude potential outliers caused by dust particles passing through the beam and false positives (when the acquisition is triggered by a transient noise spike), so that only legitimate ring-down events are taken into account."
A paragraph was added in Sect. 3.4 (see the answer to the next question of the referee 1).

Regarding "*Linear fit and orthogonal distance algorithm are mentioned in the caption. This should be discussed in the text, with appropriate references. Also include information how the fit errors were determined*":

A sentence was added in the line 9, page 15:
"…respectively. The data in the correlation plots of the $CH_3O_2$ concentrations determined by FAGE (*y*-axis) and CRDS (*x*-axis) ((Figs. 6b, 7b and 8b) were fitted using an orthogonal distance linear regression fit (Boggs et al., 1987), which accounts for errors in both the *y*- and *x*-directions. The gradient of the correlation plot at 80 mbar of He + $O_2$ (Fig. 6b)…"

Captions of figures 6, 7 and 8: "Each point is an averaged value over… " were changed to: "Each point is a value averaged over … " as suggested by the referee.

*Page 16-17, Figures 6, 7, and 8. The 80 mbar data are averaged for 3 seconds, while the 100 mbar and 1000 mbar data are averaged for 5 seconds. Suggest discussing the logic for selecting various averaging times in the text. For the intercomparison, would it not be better to average for 1 minute or more, and generate different radical concentrations by adjusting the concentrations of precursors and/or the lamp intensity, rather than use the decays to achieve different concentrations? Suggest discussing the logic of the experimental design in the body of the document.*

As explained in lines 2–4, page 15 of the MS, the comparison data were generated: (a) by delivering various concentrations of the reagents (as suggested by the referee above, so this was already done) to achieve different $[CH_3O_2]$ that decreased slowly in time due to the reagent consumption and (b) by turning off the HIRAC lamps to get a rapid decay of $[CH_3O_2]$ in time. This way the LIF method was tested in comparison with the CRDS method by monitoring both slow and rapid changes of $CH_3O_2$ concentrations and a wide range of $[CH_3O_2]$ was covered (see main manuscript). Lines 2–4, page 15 clearly explain how $[CH_3O_2]$ were generated in the comparison experiments and, hence we feel that no additional text describing this procedure is needed.

The intercomparison data were averaged over several seconds to get enough points in the rapid part of the $CH_3O_2$ kinetic decay. In order to not alter the level of noise of the data generated with the lamps on compared to the noise level of the data where the lamps were off, the same averaging time was used for all the duration of an intercomparison measurement (both periods with the lamps on and with the lamps off as shown in Figs 6a, 7a and 8a).

The paragraph below was added after line 37, page 14 to address the referee's comment regarding the difference in averaging times of the data: 3s (Fig.6) and 5 s (Figs 7 and 8).

"As the acquisition rate of CRDS (6.5 Hz in average) differed compared to the FAGE acquisition rate (in the range 1–10 Hz) the comparison data were averaged to enable comparison of $[CH_3O_2]$ by the two instruments at the same moments of time. The averaging interval of time was chosen in the range 3–5 s depending on the comparison measurement to average at least 10 ring-down events over each time interval as the CRDS data were filtered to exclude outliers caused by dust particles passing through the light beam trapped in the optical cavity and the number of encountered 'dust events' varied from one experiment to another."

*Page 17, Figure 8 caption. Suggest "The measurements by FAGE are shown in red and the measurements by CRDS are plotted in black."*

The sentence clearly refers to the comparison measurement in Fig. 8(a). As in Fig. 8(a) it is shown only one measurement there is no need to change the sentence:

"The measurement by FAGE is shown in red and the measurement by CRDS is plotted in black."

*Page 17, line 33. Suggest "...FAGE detection cell (from 3.3 to 0.9 mbar when sampling from a pressure of 1000 mbar." It is not clear why three pressures are listed, or what the "respectively" refers back to. Suggest reworking this last sentence.*

Lines 33–34, page 17 were reworded:

"…in the FAGE detection cell (from 3.3 mbar, corresponding to a total HIRAC pressure of 1000, to 0.9 mbar, corresponding to a total chamber pressure of 100 or 80 mbar)."

*Page 18-19. For future reference, it is much easier for readers if the references are formatted as "hanging" paragraphs.*

We thank to the referee for the suggestion for future publications. We would like to note that the present format of the list of the references followed the journal instructions. It is expected that during any typesetting of the MS that indenting of paragraphs as suggested by the referee will be implemented.

---

## Author Comment (AC3) · 20 Feb 2020

**Author response to anonymous referee #2 on "An intercomparison of CH$_3$O$_2$ measurements by Fluorescence Assay by Gas Expansion and Cavity Ring–Down Spectroscopy within HIRAC (Highly Instrumented Reactor for Atmospheric Chemistry)" by L. Onel et al.**

*Why did the authors measure absorption at 7488 cm-1 when Fittschen(2019) report σ (v) at 7489 cm-1? Given the results in figure 3, why do the authors consider the cross sections at the two wavelengths to be equal?*

The methane and acetone delivery led to a decrease in the ring down time due to their absorbance in the range from ~7486 to 7491 cm$^{-1}$ where the CH$_3$O$_2$ spectrum was measured. The measured absorption coefficient of acetone in a typical concentration of ~$9 \times 10^{14}$ molecule cm$^{-3}$ was practically constant, ~$8 \times 10^{-9}$ cm$^{-1}$ from ~7486 to 7491 cm$^{-1}$. However, CH$_4$ displays a more structured absorption spectrum in the probed region. Therefore, the CH$_3$O$_2$ spectrum was mapped out as a series of point measurements at fixed wavenumbers between the CH$_4$ absorption lines in the range from ~7486 to 7491 cm$^{-1}$ at 80 mbar of He + O$_2$ and 100 mbar of synthetic air (see lines 15-18, page 10). The wavenumber of 7487.98 cm$^{-1}$ was chosen for the FAGE – CRDS intercomparison measurements as there "the absorption feature is sufficiently strong and furthest in wavelength from interfering methane absorption lines…" (lines 17-18, page 10).

The following text was included in line 18, page 10:

"…was determined (Sect. 3.2). The absorption coefficient of CH$_4$ was about 7 times lower at 7487.98 cm-1 than at 7489.16 cm$^{-1}$, i.e. at the peak of the CH$_3$O$_2$ spectral feature where Fittschen (2019) reported $\sigma_{CH3O2}$. Therefore, 7487.98 cm$^{-1}$ (rounded to 7488 cm$^{-1}$ henceforth) was chosen as the measurement point instead of the value of 7489.16 cm$^{-1}$ used by Fittschen (2019). Each data point in Fig. 3…"

We do not consider that the value of the cross section, $\sigma_{CH3O2}$ at 7487.98 cm$^{-1}$ is equal to $\sigma_{CH3O2}$ at 7489.16 cm$^{-1}$ and the text does not state this. Lines 13 – 17, page 12 clearly explains the difference:

"To enable a comparison at 7487.98 cm-1 with the very recent measurement of Fittschen (2019), who found $2.20 \times 10^{-20}$ cm$^2$ molecule$^{-1}$ at 7489.16 cm$^{-1}$, $\sigma$(7487.98 cm$^{-1}$) = $1.49 \times 10^{-20}$ cm$^2$ molecule$^{-1}$ obtained in this work was multiplied by the $\sigma$(7489.16 cm$^{-1}$):$\sigma$(7487.98 cm$^{-1}$) ratio obtained by using the high resolution spectrum reported by Faragó et al. (2013) (Fig. 3). The obtained value, $\sigma$(7489.16 cm$^{-1}$) = $(1.9 \pm 0.3) \times 10^{-20}$ cm$^2$ molecule$^{-1}$ is in reasonable agreement with the result of Fittschen (2019), $\sigma$(7489.16 cm$^{-1}$) = $2.2 \times 10^{-20}$ cm$^2$ molecule$^{-1}$."

*Finally it would be helpful to have stated that the cross-section used is not the more standard integrated cross-section used by HITRAN and other databases.*

The integrated cross-section (the 'line strength') is useful when absorption lines are fitted (area under the fitted curved is proportional to line strength times concentration). For measurements at one wavelength, the 'absorption cross section' is the more appropriate, since the absorption coefficient is the concentration times the cross section. However, in case there is any confusion for those more accustomed to HITRAN, we have added in line 16, page 11:

"…, $\sigma$(7488 cm-1). Note that the cross-section used is not the more standard integrated cross-section used by HITRAN and other spectral databases. $CH_3O_2$ radicals…"

*The quantity, L, the length also requires further explanation. It should not refer to the 1.4 M value of the mirror separation in figure 2. (Morover, the comments on p14, line20-21 that one can improve the sensitivity by increasing the mirror separation, are incorrect.) The appropriate value of L is the effective path length of the sample, taking into account the diminishing concentration of CH3O2 near to and into the extensions that support the mirrors.*

As [$CH_3O_2$] was practically homogeneous across the entire length of the mirror separation the effective cavity length was considered equal to the mirror separation, $L = 1.4$ m.

The FAGE measurements of $CH_3O_2$ across the HIRAC diameter (1.2 m, 86% of the value of $L$) described in Sect.2.2.3 in the main manuscript showed that, indeed [$CH_3O_2$] was practically homogeneous across the chamber diameter. Each mirror was coupled to HIRAC by a 10 cm long system of flanges (14% from L) shown in Fig. 2 in the main manuscript. Our previous publication (Onel et al. 2017a in the manuscript references), reporting CRDS measurements of $HO_2$ performed across the HIRAC width using the same coupling system of the cavity mirrors to the chamber as in the present work, investigated the potential impact of [$HO_2$] = 0 over the two 10 cm distances between the mirrors and HIRAC. In this 'worst case scenario' the analysis found that the value for the cross section of $HO_2$ agrees within 84% with the value found by considering [$HO_2$] homogeneous along the entire $L$. We expect that the decrease in radical concentration in the proximity of the mirrors is less significant for $CH_3O_2$ than for $HO_2$ as the wall-loss for $CH_3O_2$ (upper limit of ~$10^{-5}$ s$^{-1}$) is significantly lower than the wall-loss for $HO_2$ (0.3-0.9 s$^{-1}$). Therefore, the expected very small decrease in [$CH_3O_2$] over 14% of $L$ in our experiments are thought to have a negligible impact on the value of the $CH_3O_2$ cross-section yielded by the analysis.

Following the referee's comment:"*Morover, the comments on p14, line20-21 that one can improve the sensitivity by increasing the mirror separation, are incorrect*", line 21, page 14 was rephrased to clarify more its meaning:

"The CRDS sensitivity could be further improved by mounting the cavity mirrors along the HIRAC length, which would result in a cavity of about 2 m length containing $CH_3O_2$ radicals, and, hence above the current 1.4 m length…"

---

## Author Comment (AC4) · 20 Feb 2020

**Author response to anonymous referee #3 on "An intercomparison of CH$_3$O$_2$ measurements by Fluorescence Assay by Gas Expansion and Cavity Ring–Down Spectroscopy within HIRAC (Highly Instrumented Reactor for Atmospheric Chemistry)" by L. Onel et al.**

***Minor comments:***
*P2-P3: Several techniques are discussed for the measurement of peroxy radicals. The authors should also briefly discuss the use of chemical ionization mass spectrometry methods such as published in Noziere and Hanson (2017), Noziere and Vereecken (2019), Hansel et al. (2018), Jokinen et al. (2014), etc.*

We thank the referee for pointing out these additional references to include in the MS. We have now extended the introduction to discuss briefly the use of CIMS for the detection of speciated RO$_2$ radicals. The following text has been added to the MS after line 34, page 2:

"CIMS methods using reagent ions such as H$_3$O$^+$(H$_2$O)n, NO$_3^-$ and NH$_4^+$ have been employed in the simultaneous and selective detection of RO$_2$ in a number of recent studies (Noziere and Hanson, 2017; Noziere and Vereecken, 2019; Hansel et al., 2018; Jokinen et al., 2014). Volatile small RO$_2$ radicals such as CH$_3$O$_2$ have been selectively measured in CIMS laboratory experiments with detection limits between $\sim$1 $\times$ 10$^8$ –1 $\times$ 10$^9$ molecule cm$^{-3}$ (Noziere and Hanson, 2017; Noziere and Vereecken, 2019). CIMS with NO$_3^-$ reagent ion has been employed in field measurements to record diurnal profiles of some highly oxygenated low–vapour pressure RO$_2$ radicals produced in the ozonolysis of monoterpenes peaking at a few 10$^7$ molecule cm$^{-3}$ (Jokinen et al., 2014). "

*P4 L16 & P5 L19: CH4 is used during calibration experiments as a precursor for CH3O2 and is added in the Water-photolysis calibrator and the HIRAC chamber at concentrations as high as 2.5E17 molecule/cm3. Can the authors comment on the potential impact of CH4 on the quenching of CH3O in the detection cell?*

The answer to this question is given in the response to the first specific question asked by referee 1, where no quenching effect of the CH$_3$O($A$) fluorescence by CH$_4$ was found in the present experiments. Experiments were performed at several [CH$_4$] and no difference in the sensitivity factor for CH$_3$O$_2$ was observed. At the small mixing ratios of CH$_4$ used, and following expansion to low pressure in the FAGE fluorescence chamber, the quenching of CH$_3$O(A) by CH$_4$ is expected to be very minor compared with that of O$_2$ or N$_2$.

*P7 Eq. 6: Please define kloss*

An explanation about kloss were added in lines 4–5, page 7:

"…the potential for a loss of CH$_3$O$_2$ to the walls was investigated. As circulation fans were used during all the experiments, the 'movement' of CH$_3$O$_2$ radicals within the chamber is in part molecular diffusion and in part convection. Therefore, the parameter $k_{loss}$ is controlled by both convection and diffusion processes. By incorporating the wall loss…"

*P6 L9-13 & P12 L4-6: What were the fitted values for kloss? Are the values inferred from the two experiments consistent with each other?*

FAGE was sampling from a point close to the chamber centre while CRDS measured $CH_3O_2$ right across the HIRAC diameter (Fig. 2 in the main text). However, the kinetic decay analysis demonstrated that wall losses were negligible in both FAGE and CRDS measurements.
The analysis of the kinetic decays monitored by the two instruments has been done in the same way for both $[CH_3O_2]_{FAGE}$ and $[CH_3O_2]_{FAGE}$ decays. The upper limit for $k_{loss}$ in the FAGE measurements was added to the text as shown in the answer to the first referee's comments:

"… the small values extracted for $k_{loss}$ (upper limit of ~ $1 \times 10^{-5}$ $s^{-1}$) fitting Eq. (6) to the FAGE data demonstrates that wall losses can be neglected…" was added on page 7, lines 11-12."

The same upper limit was obtained for $k_{loss}$ by analysing the kinetic decays measured by CRDS. The result was added in line 6, page 12:

"…are statistical uncertainties. The values extracted for $k_{loss}$ by fitting Eq. (9) to the CRDS data were small and similar to the values obtained by fitting Eq. (6) to the kinetic decays monitored by FAGE. An upper limit of ~ $1 \times 10^{-5}$ $s^{-1}$ was obtained for $k_{loss}$ in both FAGE and CRDS measurements, showing that wall losses are negligible. From fitting…"

*P6 L22: "1E-10" should read "1E-9"*
"1E-10" was corrected to "1E-9"

*P15 L9-12 & L17-18 & L21-22: The authors show that correlation plots between FAGE and CRDS exhibit slopes that are close to unity. However, the y-intercepts of the regression lines are not discussed. Were the intercepts not statistically significant?*

The *y*-intercepts of the FAGE – CRDS correlation plots (Figs. 6b, 7b and 8b) have either a small negative value (Fig. 6b) or a positive value (Fig. 7b and 8b). We believe that the main source for the *y*-intercept values derived by the linear fit to the data is the method used to determine the background of the CRDS measurements. The background ring-down time (the ring-down time in the $CH_3O_2$ absence, $\tau_0$) increased slightly during the time intervals with the lamps on due to the slow depletion of the reagents (methane or acetone). However, $\tau_0$ could not be measured simultaneously with the ring-down time in the presence of the $CH_3O_2$ radicals, $\tau$. Therefore, the background was regularly monitored by turning the lamps off, as explained in lines 9 – 10, page 10:

"As it is not possible to measure $\tau_0$ and $\tau$ simultaneously, the background was monitored regularly during each experiment by switching off the photolysis lamps and allowing the signal to return to the baseline."

We also added a new paragraph describing the impact of the acetone and methane absorption on $\tau_0$ to the main text (see the answer to the first specific comment of the referee 1).

The background in the FAGE measurements could not also be recorded simultaneously with the $CH_3O_2$ FAGE signal as it required the FAGE instrument measured off-line. Therefore, the off-line measurement was taken at the end of each on-line measurement. However, the FAGE measurement background was independent on any changes in the composition of the HIRAC gas mixture. Therefore, we believe that the source of the linear regression intercepts mentioned by the referee comes from the uncertainties associated with the determination of the CRDS measurement background as explained above. The intercepts are not significant as their values are only a few percent from the largest $[CH_3O_2]$ shown in each correlation plot.

*P16 Figures 6-7: When the lamps are turned off, (1) the CRDS measurements seem to decrease to lower values than FAGE and (2) the FAGE measurements seem to reach a plateau more rapidly than the CRDS. Could the authors comment on this?*

As it was not possible to measure $\tau_0$ and $\tau$ simultaneously the background ring-down time was recorded regularly by turning off the chamber lamps to account for the slow decrease in the reagent (methane or acetone) concentrations (*vide supra*). The method led to typical small deviations of the baseline of the CRDS kinetic decays from zero and to the slight differences between the baselines of the $[CH_3O_2]_{FAGE}$ decay and $[CH_3O_2]_{CRDS}$ decay mentioned by the referee.

The two referee's observations are coupled to each other – the $[CH_3O_2]_{CRDS}$ continuing to go down, and $[CH_3O_2]_{FAGE}$ levelling off quicker. The FAGE measurement levelling is due to the second order kinetics going to a constant value, $[CH_3O_2] = 0$ whereas $[CH_3O_2]_{CRDS}$ continues to go down as some reaction products absorbing at the measuring wavenumber ($7488 \text{ cm}^{-1}$) are changing with time. This in turn slightly changes the absorption even when the $CH_3O_2$ has reached zero concentration. Both these effects are more evident at 80 and 100 mbar than at 1000 mbar. The better FAGE – CRDS agreement at longer times at 1000 mbar than at 80 and 100 mbar could be due to a greater wall loss of the absorbing products at reduced pressures, where diffusion becomes more significant. However, even at 80 and 100 mbar the discrepancies noticed by the referee are minor and the FAGE – CRDS correlation plots, which incorporate all the temporal decay data show a good agreement under all conditions.